# Cortical RORβ is required for layer 4 transcriptional identity and barrel integrity

Erin A Clark[1]*, Michael Rutlin[1], Lucia S Capano[1], Samuel Aviles[1], Jordan R Saadon[1], Praveen Taneja[1], Qiyu Zhang[1], James B Bullis[1], Timothy Lauer[1], Emma Myers[1], Anton Schulmann[2], Douglas Forrest[3], Sacha B Nelson[1]*

[1]Department of Biology and Program in Neuroscience, Brandeis University, Waltham, United States; [2]Janelia Research Campus, Ashburn, United States; [3]Laboratory of Endocrinology and Receptor Biology, National Institutes of Health, NIDDK, Bethesda, United States

**Abstract** Retinoic acid-related orphan receptor beta (RORβ) is a transcription factor (TF) and marker of layer 4 (L4) neurons, which are distinctive both in transcriptional identity and the ability to form aggregates such as barrels in rodent somatosensory cortex. However, the relationship between transcriptional identity and L4 cytoarchitecture is largely unknown. We find RORβ is required in the cortex for L4 aggregation into barrels and thalamocortical afferent (TCA) segregation. Interestingly, barrel organization also degrades with age in wildtype mice. Loss of RORβ delays excitatory input and disrupts gene expression and chromatin accessibility, with down-regulation of L4 and up-regulation of L5 genes, suggesting a disruption in cellular specification. Expression and binding site accessibility change for many other TFs, including closure of neurodevelopmental TF binding sites and increased expression and binding capacity of activity-regulated TFs. Lastly, a putative target of RORβ, *Thsd7a*, is down-regulated without RORβ, and *Thsd7a* knock-out alone disrupts TCA organization in adult barrels.

*For correspondence:
eaclark@brandeis.edu (EAC);
nelson@brandeis.edu (SBN)

## Introduction

Localization of function is a fundamental principle organizing mammalian brain circuitry. Structure to function mapping is particularly striking in sensory input to L4 of the neocortex (*Woolsey and Van der Loos, 1970*; *Catania and Kaas, 1995*). L4 neurons are distinctive in their propensity to form cellular aggregates, or modules, that receive segregated thalamic inputs and represent features of the sensory periphery. Whisker barrels in the rodent somatosensory cortex are a prototypical example, but other somatosensory modules within L4 are also present in the cortices of insectivores, carnivores and primates (*Krubitzer and Seelke, 2012*), and columns receiving segregated input are present in the visual cortices of carnivores and primates, and in other cortical regions (*Mountcastle, 1997*). At the same time, gene expression studies in mouse and human show that L4 neurons also have a distinctive transcriptional identity that includes expression of RORβ (*Zeng et al., 2012*). Despite these two striking features, little is known about the relationships between transcriptional identity, the mechanisms that establish and regulate that identity, and features of L4 cytoarchitecture.

Researchers have long used the rodent whisker pathway to study cytoarchitecture development (*Hand and Strick, 1982*; *Fox, 1992*; *Yang et al., 2018*). The whisker map is organized into microcolumnar units called barrels located in primary somatosensory cortex (S1). In mice, L4 cortical neurons assemble into columns that form barrel walls and input is relayed via thalamocortical afferents

(TCAs), which cluster in the center of barrel hollows. Each whisker is projected through corollary maps in the brainstem and ventrobasal thalamus (*Van Der Loos, 1976*) before reaching S1.

Many proteins and pathways are required for presynaptic organization of TCAs and/or postsynaptic organization in L4 (*Li and Crair, 2011*; *Wu et al., 2011*; *Erzurumlu and Gaspar, 2012*). Much of what we know has focused on the requirement of input activity and intact signaling pathways. Genetic disruption of synaptic transmission via glutamate (*Iwasato et al., 1997*; *Iwasato et al., 2000*; *Hannan et al., 2001*; *Datwani et al., 2002*; *Li et al., 2013*; *Ballester-Rosado et al., 2016*), or serotonin pathways (*Cases et al., 1995*; *Salichon et al., 2001*) perturb some aspect of barrel organization. Several related signal transduction pathways are also required (*Abdel-Majid et al., 1998*; *Barnett et al., 2006*; *Inan et al., 2006*; *Watson et al., 2006*; *Lush et al., 2008*).

Barrel formation is also regulated transcriptionally. Transcription factors (TFs) such as Bhlhe22/ Bhlhb5 and Eomes are involved in the early stages of cortical arealization and barrel development (*Arnold et al., 2008*; *Joshi et al., 2008*; *Elsen et al., 2013*). Downstream of these early developmental processes activity-dependent TFs, including Lmo4, NeuroD2, and Btbd3 regulate aspects of barrel organization in response to TCA inputs (*Ince-Dunn et al., 2006*; *Kashani et al., 2006*; *Huang et al., 2009*; *Matsui et al., 2013*). In addition, the TFs retinoic acid-related orphan receptor alpha (RORα) and beta (RORβ), are also implicated in barrel formation. RORα and RORβ are expressed in regions of the somatosensory barrel map, with RORα expressed in brainstem, thalamus and cortex, and RORβ in thalamus and cortex (*Nakagawa and O'Leary, 2003*). Recently, RORα was shown to be required in the thalamus and cortex for proper TCA segregation and barrel wall formation (*Vitalis et al., 2018*). Mis-expression of RORβ in neocortex is sufficient to drive cortical neuron clustering and TCA recruitment to ectopic barrel-like structures (*Jabaudon et al., 2012*). Together these studies have identified multiple TFs with major roles in early barrel development that likely set the stage for more downstream terminal differentiation TFs and activity-regulated TFs to hone the network. Early cortical development, TCA pathfinding, and activity dependent gene regulation are prolific areas of research. However, the later stages of neuronal specification and the molecular mechanisms of TFs involved in barrel development are currently underexplored. TFs such as Bhlh5 and Eomes have broad roles and are widely expressed in the cortex while the more narrowly expressed TFs such as Btbd3 are downstream of activity input leaving a gap in our understanding of the intermediate steps that connect cortical development to activity driven processes. Given the restricted layer-specific expression of RORβ and its up-regulation concomitant with the final stages of barrel formation and the onset of input activity, we hypothesized it would be a good candidate to study transcriptional mechanisms connecting cellular specification in L4 with cytoarchitecture and network development.

We show that in addition to being sufficient, RORβ is also required for both pre- and postsynaptic barrel organization. Without RORβ in the cortex, L4 neurons fail to migrate tangentially and do not organize into barrel wall structures. This also reduced TCA segregation shortly after barrel formation would have normally occurred. Interestingly, TCA segregation also declined as animals aged. Without RORβ, L4 gene expression and chromatin accessibility were disrupted, with L4-specific genes down-regulated and L5-specific genes up-regulated suggesting a disruption in terminal cellular identity. This involved complex changes in the expression and/or chromatin accessibility at binding motifs for many TFs in addition to RORβ, including developmental regulators and activity-regulated TFs. L4 neurons also received delayed excitatory input, a key step in barrel development. Lastly, we identify a putative direct gene target of RORβ, *Thsd7a*, that is down-regulated without RORβ and is required for maintained TCA organization in adulthood. Together these data characterize the role of RORβ across multiple levels to connect molecular and transcriptional mechanisms to cortical organization and place RORβ as a key regulator of a complex developmental transition orchestrating terminal L4 specification and initiating activity responsiveness.

## Results

Cortical barrels in mice are complex structures. Cell-sparse barrel hollows are where thalamic projections are concentrated. Barrel walls are formed by cortical cell aggregates that surround the TCAs. Barrel septa consist of the intermediate spaces between barrel walls (*Woolsey and Van der Loos, 1970*). To assess the impact of RORβ loss on barrel organization we used two staining methods. Barrel walls were visualized by Nissl staining (*Van der Loos and Woolsey, 1973*) and barrel hollows

were visualized by vesicular glutamate transporter 2 (VGLUT2), which is strongly expressed in TCAs (*Fremeau et al., 2001*; *Liguz-Lecznar and Skangiel-Kramska, 2007*), or as clusters of reporter expressing afferents from VPM neurons. This strategy allowed clear identification of changes in either structure independently. Cytochrome oxidase (CO) staining was also used in some conditions, but the presence of CO signal in both barrel walls and TCAs made it less useful.

## RORβ is required for postnatal barrel wall formation and influences segregation of thalamocortical afferents (TCAs)

To begin exploring RORβ function in barrel organization, we used a global, constitutive knock-out (KO), which contains a GFP expression cassette knocked-in to the *Rorb* locus. *Rorb*$^{GFP/+}$ mice express GFP in RORβ expressing cells allowing identification of barrel cortex without significant disruption to barrel structures or neuronal function (*Liu et al., 2013*). *Rorb*$^{GFP/+}$ mice were used as controls (Ctl), while *Rorb*$^{GFP/GFP}$ mice disrupt both copies of *Rorb* to generate a KO. Controls showed no detectable disruption to barrel organization compared to WT animals (*Figures 1A* and *2A*).

Barrels form around postnatal day 5 (*Rice and Van der Loos, 1977*). Nissl staining of barrel walls at P7, P30, and P60 showed that RORβ is required for barrel wall formation. Representative images of Nissl and GFP are shown in *Figure 1A* where the lack of barrel wall organization is clearly visible at P7 and remains disrupted at P30. *Figure 1B* quantifies this effect as the contrast between barrel hollows and barrel wall/septa fluorescence intensity. Contrast was calculated as (barrel - septa) / (barrel + septa) where septa includes barrel walls (see methods for details). Quantification demonstrated a near complete lack of contrast in KO barrel cortex supporting a lack of cortical organization.

While TCAs have been shown to instruct cortical cell organization we hypothesized the lack of barrel walls might reciprocally affect TCA organization. TCAs visualized by VGLUT2 staining showed an intact pattern of barrel hollows at P7 in KO animals, *Figure 2A*. However, careful quantification of the VGLUT2 contrast between hollows and septa showed a significant decrease in the KO suggesting loss of RORβ and/or the lack of barrel walls had a mild but measurable effect on TCA segregation. Interestingly, as animals aged into adulthood TCA segregation also declined in control as well as *Rorb* KO animals. Disorganization in the *Rorb* KO was characterized by both loss of quantifiable VGLUT2 contrast as well as the qualitative barrel patterning most obvious at P60 between Ctl and KO in *Figure 2A*. Both genotype and age significantly affected VGLUT2 contrast (genotype p=4.5e-07 and age p=2.6e-06 by two-way ANOVA) but did not interact significantly. Comparing pairwise across ages we find a significant decline in TCA organization between P7 and P20 controls, with no significant change from P20 to P60. This suggests that while both age and loss of RORβ significantly reduced contrast, loss of RORβ did not significantly change the time course of TCA desegregation.

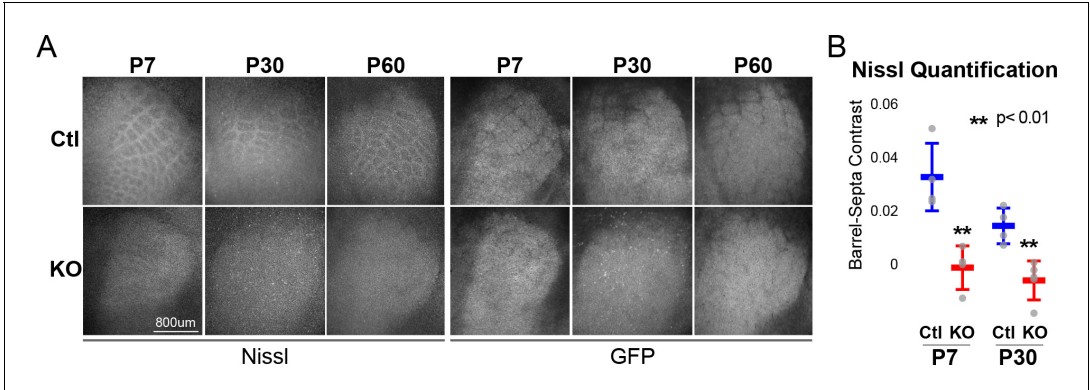

**Figure 1.** RORβ is required for postnatal barrel wall formation. Nissl staining on tangential sections of flattened cortices after global, constitutive knock-out shows barrel wall organization requires RORβ. (**A**) Nissl staining (Left) in whisker barrel field as identified by strong GFP expression (Right). Control (Ctl) and *Rorb* knock-out (KO) animals were age matched at P7, P30, and P60. (**B**) Quantification of barrel hollow to barrel walls/septa contrast (Barrel-Septa Contrast) from Nissl staining. N = 4 age-matched animals for each genotype (Ctl or KO). Two tissue sections containing the largest portions of whisker barrel field identified by GFP signal were averaged per animal. Whisker plots show the median per animal ± standard deviation. Gray points show mean contrast for each animal. P-value by independent sample t-test, between Ctl and KO at each timepoint.

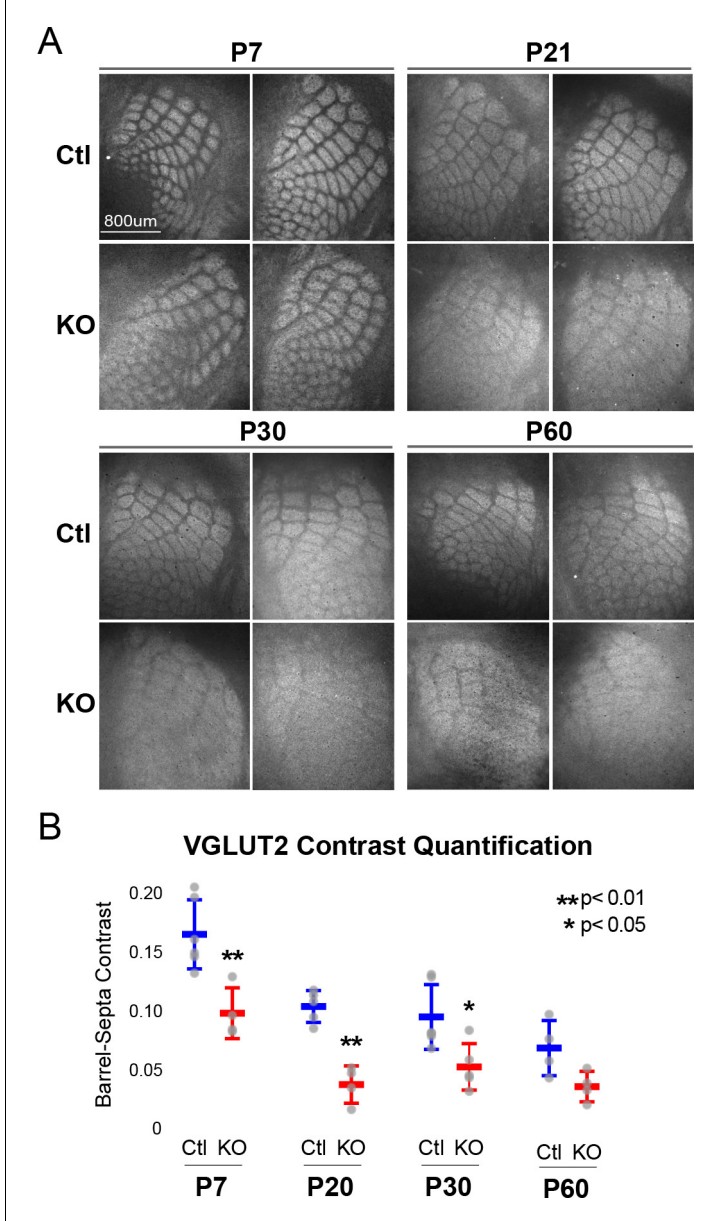

**Figure 2.** *Rorb* KO reduces thalamocortical afferent (TCA) segregation. (**A**) VGLUT2 staining of excitatory thalamic axon terminals in cortical whisker barrels shows normal initial TCA patterning at P7 but with reduced barrel-septa contrast in *Rorb* KO, and further reductions in contrast with age in both KO and Ctl. Ctl and *Rorb* KO animals were age matched. (**B**) Quantification of barrel hollow to barrel walls/septa contrast (Barrel-Septa Contrast) in VGLUT2. N = 4–6 age-matched animals for each genotype (Ctl or KO; each section shown is from a separate animal). Two tissue sections containing the largest portions of whisker barrel field identified by GFP signal were averaged per animal. Whisker plots show median contrast per animal ± standard deviation. Gray points show mean contrast for each animal. P-value by independent sample t-test, between Ctl and KO at each timepoint.

The online version of this article includes the following figure supplement(s) for figure 2:

**Figure supplement 1.** Barrel-septa contrast of VPM-specific afferents.

---

To examine whether loss of VGLUT2 contrast could be due to late arrival of VGLUT2[+] inputs from outside the VPM we injected AAV expressing mCherry under the hSyn promoter specifically into the VPM (*Figure 2—figure supplement 1A*). The VGLUT2 barrel-septa contrast was comparable to the barrel-septa contrast in the VPM-specific mCherry filled afferents at P30 strongly suggesting loss of VGLUT2 contrast with age is due to loss of TCA organization (*Figure 2—figure supplement 1B-C*).

Together these data show that RORβ is critical for normal whisker barrel formation and, loss of TCA segregation into adulthood suggests that time/age continues to affect cytoarchitecture.

## RORβ is required in the cortex but not the thalamus for barrel organization

In addition to L4 excitatory neurons, RORβ is expressed in the thalamic neurons that project to barrel hollows. To assess whether the disruption of barrels is dependent on RORβ expression in thalamus and/or locally in cortex we used a floxed allele of *Rorb* (*Rorb*$^{f/f}$) crossed to Cre driver lines generating tissue-specific disruption of RORβ as diagrammed in *Figure 3A*. A knock-in line expressing *Cre* from the serotonin transporter gene, *Sert* (*Slc6a4* or 5-HTT) locus was used to knock-out *Rorb* in the thalamus. The *Sert*$^{Cre}$ line alone showed a mild disruption to TCA organization without disrupting barrel walls, suggesting the *Cre* knock-in might be hypomorphic (*Figure 3B–C*). However, thalamic KO of *Rorb* (*Sert*$^{Cre}$ *Rorb*$^{f/f}$) showed no additional disruption to TCAs or barrel walls. This is consistent with the observation that *Rorb* KO also did not disrupt barreloid organization (*Figure 3—figure supplement 1A*). Thus, loss of RORβ in thalamic neurons was not responsible for the loss of cortical wall organization or the majority of TCA disorganization observed in the global *Rorb*$^{GFP/GFP}$ KO.

A knock-in line expressing *Cre* from the *Emx1* locus removed RORβ specifically in forebrain structures. *Emx1*$^{Cre}$ alone showed no significant disruption to barrel organization (*Figure 3D–E*). However, barrel organization was significantly disrupted by cortical KO of *Rorb* (*Emx1*$^{Cre}$ *Rorb*$^{f/f}$). In addition, a *CamK2a*$^{Cre}$ diver line that removes RORβ in the cortex after barrel formation, showed no effect. *CamK2a*$^{Cre}$ activated expression of a tdTomato reporter from the *Rosa26* locus in only a subset of GFP$^+$ L4 neurons (*Figure 3—figure supplement 1B*), therefore it is not clear whether late expression of RORβ is expendable or whether expression in a subset of L4 neurons is sufficient for barrel organization. Together these data demonstrate that RORβ is required in the cortex prior to barrel formation. Loss of RORβ in the thalamus does not disrupt barrel architecture, suggesting RORβ drives barrel wall organization through cell-intrinsic mechanisms within layer 4.

## RORβ is required for expression of a layer four gene profile and repression of layer five genes

Because RORβ is a transcription factor we hypothesized loss of function would change gene expression in L4 neurons. To test this, RNA-seq was performed on sorted GFP$^+$ cells from micro-dissected L4 S1. We were careful in this dissection to exclude a small population of GFP$^+$ L5 neurons. Differential expression analysis between *Rorb*$^{GFP/+}$ and *Rorb*$^{GFP/GFP}$ cells identified many dysregulated genes (fold change ≥2, adjusted p-value<0.01). At postnatal day 2 (P2) and prior to barrel formation, 246 genes were significantly disrupted with 51% down-regulated in the KO. At P7, just after barrel formation, 433 genes were disrupted with 36% down-regulated. At P30, 286 genes were disrupted with 37% down-regulated. Examining the overlap between ages we find very few genes significantly disrupted in the same direction across time points, suggesting highly dynamic and complex regulation, *Figure 4A, B*.

RORβ expression is a key feature distinguishing L4 neurons (*Lein et al., 2007*). To examine the effect of RORβ loss on layer-specific transcriptional identity we assessed the layer specificity of genes differentially expressed between control and *Rorb* KO (DEGs). The Allen Brain Atlas was used to manually screen all DEGs for layer-specific expression in the neocortex. Genes were considered layer-specific if the in-situ hybridization (ISH) signal appeared at least three-fold higher in one layer (considering layers 2 and 3 together). Many genes had complex specificities showing enrichment in two or more layers. These were not included for simplicity. Grouping DEGs based on the layer they are normally expressed within, we see that DEGs which should be expressed in upper layers were generally down-regulated and DEGs that should be expressed in deep layers were generally up-regulated in the *Rorb* KO, *Figure 4A–D*. The strongest effects were loss of many L4 genes and increased expression of many L5 genes. While many L4 and L5 genes were affected, this was not a global identity switch. Many L4 and L5 genes identified from the Allen Brain Atlas were not differentially expressed. In order to assess the statistical significance of the down-regulation of L4 genes and up-regulation of L5 genes we used the Allen Atlas differential search function to contrast L4 to L5 of primary somatosensory cortex (SSp) and included all genes with >1.5 fold change and expression threshold >1.6 (*Figure 4E–F*). Of the 102 L4-specific genes 26% were down-regulated in the KO, a

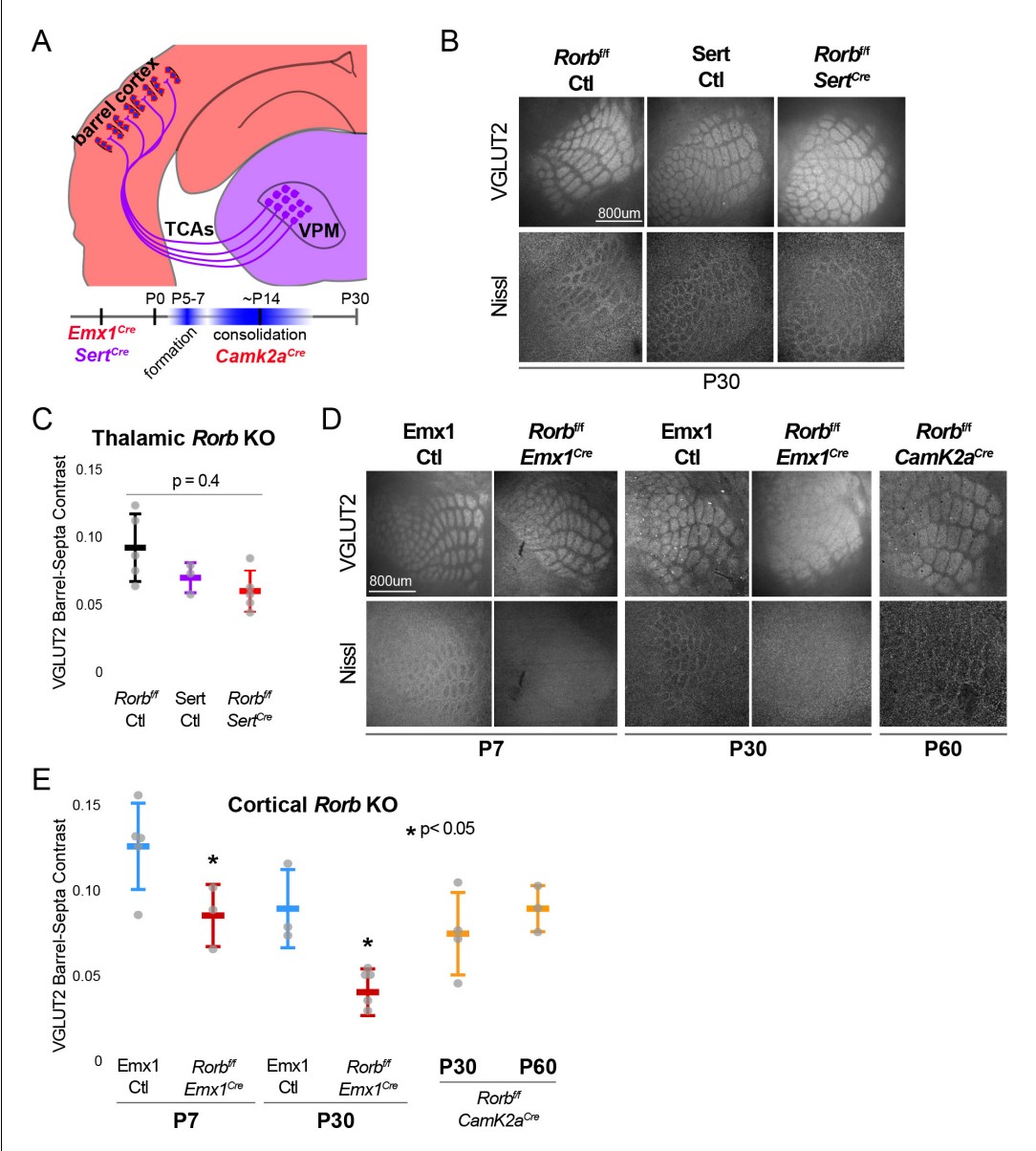

**Figure 3.** RORβ is required in the cortex but not the thalamus for barrel organization. (**A**) Diagram and timeline of Cre driver line tissue-specific expression in cortex versus thalamus and timing relative to barrel formation and consolidation. Color indicates expression in cortex (red) or thalamus (purple). (**B**) VGLUT2 and Nissl staining of whisker barrel cortex at P30 from floxed *Rorb* control without *Cre* (*Rorb*^f/f Ctl), *Sert*^Cre control (Sert Ctl) without floxed *Rorb* and the cross (*Rorb*^f/f *Sert*^Cre), which knocks out *Rorb* specifically in thalamus during embryonic development. Whisker plots as described for *Figure 1B*. (**C**) Quantification of VGLUT2 Barrel-Septa Contrast in genetic lines from B. N = 3–5 P30 animals. Quantification and plotting as described in *Figure 2B*. P-value by ANOVA. (**D**) VGLUT2 and Nissl staining of whisker barrel cortex from *Emx1*^Cre control (Emx1 Ctl) without floxed *Rorb*, and the cross (*Rorb*^f/f *Emx1*^Cre) from P7 and P30 animals, and a P60 animal from floxed *Rorb* crossed to a *CamK2a*^Cre driver line. *Emx1*^Cre knocks out *Rorb* specifically in forebrain during embryonic development, and *CamK2a*^Cre knocks out *Rorb* in forebrain neurons at postnatal weeks 2–3. (**E**) Quantification of VGLUT2 Barrel-Septa Contrast in genetic lines from D. N = 3–5 animals per age group. Quantification and plotting as described in *Figure 2B*. P-values by independent sample t-test, between Ctl and KO at each time point. *CamK2a*^Cre showed no difference from *Rorb*^f/f Ctl. Whisker plots as described for *Figure 1B*.

The online version of this article includes the following figure supplement(s) for figure 3:

**Figure supplement 1.** Reporter expression in L4 of *Camk2a*^Cre *Rorb* KO.

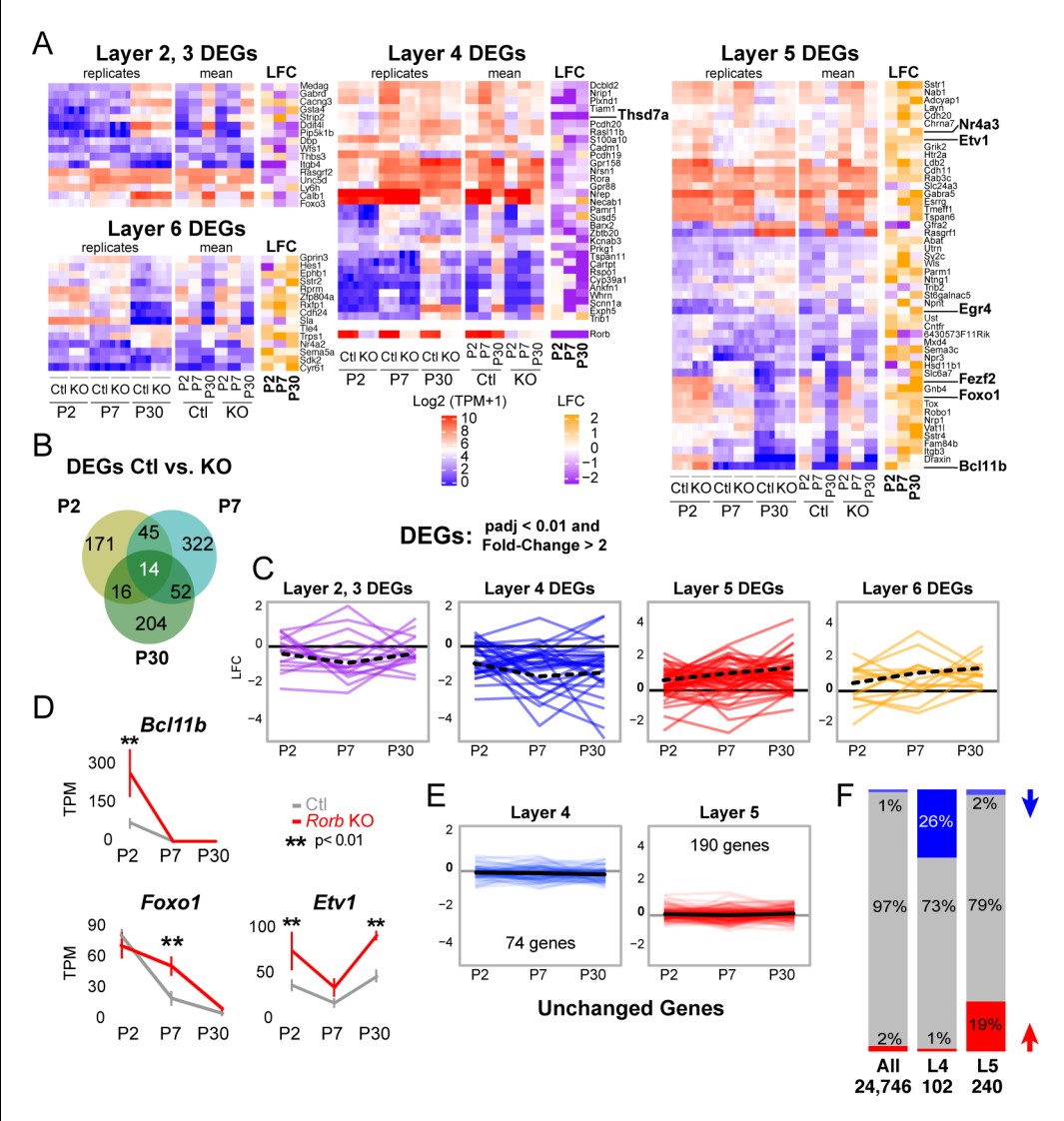

**Figure 4.** *Rorb* KO disrupts the layer four expression profile including up-regulating many deep layer genes. (A) Heatmaps showing marker genes or genes strongly enriched, as identified in the Allen Brain Atlas, for each layer of the neocortex differentially expressed between control and *Rorb* KO. Log-transformed transcripts per million (TPM) are color scaled in red and blue for each of the four RNA-seq replicates in the left most heatmap and the mean for each time point and genotype in the middle heatmaps. Log fold change (LFC) between control (Ctl) and *Rorb* KO is color-scaled in orange and purple in the right most heatmaps. (B) Numbers of differentially expressed genes (DEGs) for the three ages examined. (C) Line plots showing LFC for the same genes. The solid black line indicates no change. Negative LFC indicates decreased expression in *Rorb* KO, and LFC >0 indicates increased expression in *Rorb* KO. Each colored line is a layer-specific DEG and the dashed black line plots the mean across the group of genes. (D) RNA-seq expression of layer 5 TFs. Lines plot the mean ± SE. P by moderated t-test adjusted for multiple comparisons (Benjamini-Hochberg). (E) Additional L4 and L5 genes were identified using the Allen Brain Atlas differential search contrasting L4 SSp structures to L5 SSp. Genes with >1.5 fold change and expression threshold >1.6 were selected. Genes already shown in A-C were removed. Hence each gene shown does not meet statistical criteria for differential expression in Ctl/KO by RNA-seq. Line plots show RNA-seq LFC for each layer-specific gene. The solid black line is the mean across genes and the solid gray line indicates no change. Negative LFC indicates decreased expression in *Rorb* KO, and LFC >0 indicates increased expression in *Rorb* KO. (F) Overall (first bar), 1% of genes were downregulated (blue) and 2% were upregulated (red). Downregulated genes were overrepresented (26%) among the 102 L4-specific genes (middle bar), while upregulated genes were overrepresented (19%) among the 240 L5-specific genes. Both overrepresentations were significant (p<2.2e-16) by fisher exact test.

single gene was up-regulated, and the remainder were unchanged. Conversely, up-regulated genes were overrepresented (19%) among the 240 L5-specific genes, and a fisher exact test revealed that these overrepresentations were highly significant (p<2.2e-16). Thus, although only a portion of the L4 gene expression profile is altered by loss of *Rorb*, it is disproportionately weighted towards down-regulation of L4 genes and upregulation of L5 genes.

Several L5 genes are worth noting. Bcl11B/Ctip2, is a marker of thick-tufted L5B-type neurons and significantly up-regulated at P2 in the KO, but silenced at P7 and P30 similar to control (*Figure 4D*). Fezf2, another L5B marker and regulator of *Bcl11B* (*Chen et al., 2005*), was similarly silenced over barrel development, but was overexpressed at P30 in the KO. Foxo1 is mainly expressed in L5 at younger ages (Allen Developing Mouse Brain Atlas) declining over barrel development, but in the KO was significantly overexpressed at P7. *Etv1*, also a L5A marker (*Doyle et al., 2008*), was upregulated in the KO at both P2 and P30. Lastly, *Egr4* was up-regulated at P30 in the KO, and has been associated with *Etv1* expressing neurons (*Doyle et al., 2008*). RNAscope (*Wang et al., 2012*) in situ analysis against two L5 genes confirmed up-regulation in L4 (*Figure 5*, *Figure 5—figure supplement 1A*). Together these data support a disorganized partial shift in layer identity with many different factors implicated at distinct time points.

### *Rorb* KO disrupts transcription factor binding sites near DEGs

RORβ, Bcl11b, Foxo1, Etv1, and Egr4 are TFs that often regulate gene expression by binding to distal regulatory sites such as enhancers. There are many chromatin features of enhancers, one of which is that they are open and accessible to enzymatic fragmentation in assays such as the Assay for Transposase Accessible Chromatin (ATAC) (*Buenrostro et al., 2015*). To begin examining mechanisms involved in changing gene expression, we performed ATAC-seq on sorted GFP$^+$ L4 neurons from control and *Rorb* KO animals at P30 (*Figure 6A*). High confidence ATAC-seq peaks were assessed for differential accessibility between control and KO samples. We identified 5210 peaks with $\geq$2 fold change in accessibility (FDR < 0.02). Nearly 4-times as many regions lost accessibility (N = 4123 closed) than increased (N = 1087 opened), (*Figure 6—figure supplement 1A*). Differential ATAC peaks were primarily located in introns and intergenic regions *Figure 6—figure supplement 1B* suggesting loss of RORβ function resulted in closure of many more regulatory regions than opening.

We hypothesized that many of the closed regions might contain a RORβ binding motif while regions that opened may have binding potential for other TFs. To assess this possibility, two software algorithms (MEME and HOMER) were used to identify de novo enriched motifs from the DNA sequences of differential ATAC peaks separating closed and opened regions. This unbiased analysis also identifies which enriched sequences match known TF binding motifs. RORβ was the top motif from closed regions, *Figure 6—figure supplement 1C*. Considering only expressed TFs, the potent neurogenic factors NeuroD1 and Ascl1 were also among the top motifs in closed regions. In regions that opened, the top motifs from expressed TFs were Nfil3, Hlf, Jun, Fos, Trps1, Mef2a/c/d and Irf2. Similar analysis was performed on ATAC peaks near up or down-regulated DEGs as well as L4 and L5 DEGs. To confirm enrichment and identify motif locations we used MEME FIMO and HOMER to scan for instances of a given set of motifs. This was done for all expressed TFs either enriched in the de novo motif analysis or differentially expressed, for which high quality motif models existed. Motif instances were cross-validated by retaining only those found by both MEME and HOMER. *Figure 6B* plots the odds ratio of motifs significantly enriched compared to control regions. Many of the motifs found by de novo analysis were confirmed, including RORβ in regions that closed.

To assess which TFs might play a significant role in up or down-regulation of DEGs we varied a distance window around the transcription start site (TSS) to identify nearby ATAC or control regions containing a DNA motif. We tested for enrichment of motifs in ATAC regions near DEGs compared to motifs in control regions. We also tested whether DEGs with a nearby motif were significantly enriched compared to a control group of genes that did not change expression in the *Rorb* KO. In essence, we tested whether motifs were enriched around certain DEGs and whether a significant portion of those DEGs had a nearby motif. To reduce false positives, only motifs with significant enrichment in both tests are shown in *Figure 6C–D*.

Genes down-regulated at P30 showed significant enrichment of nearby RORβ motifs suggesting RORβ is important for gene activation (*Figure 6C*). Motifs for Nr4a1 and Nfil3 were enriched near up-regulated DEGs at P2 and P7 respectively consistent with an early role for these TFs in activating

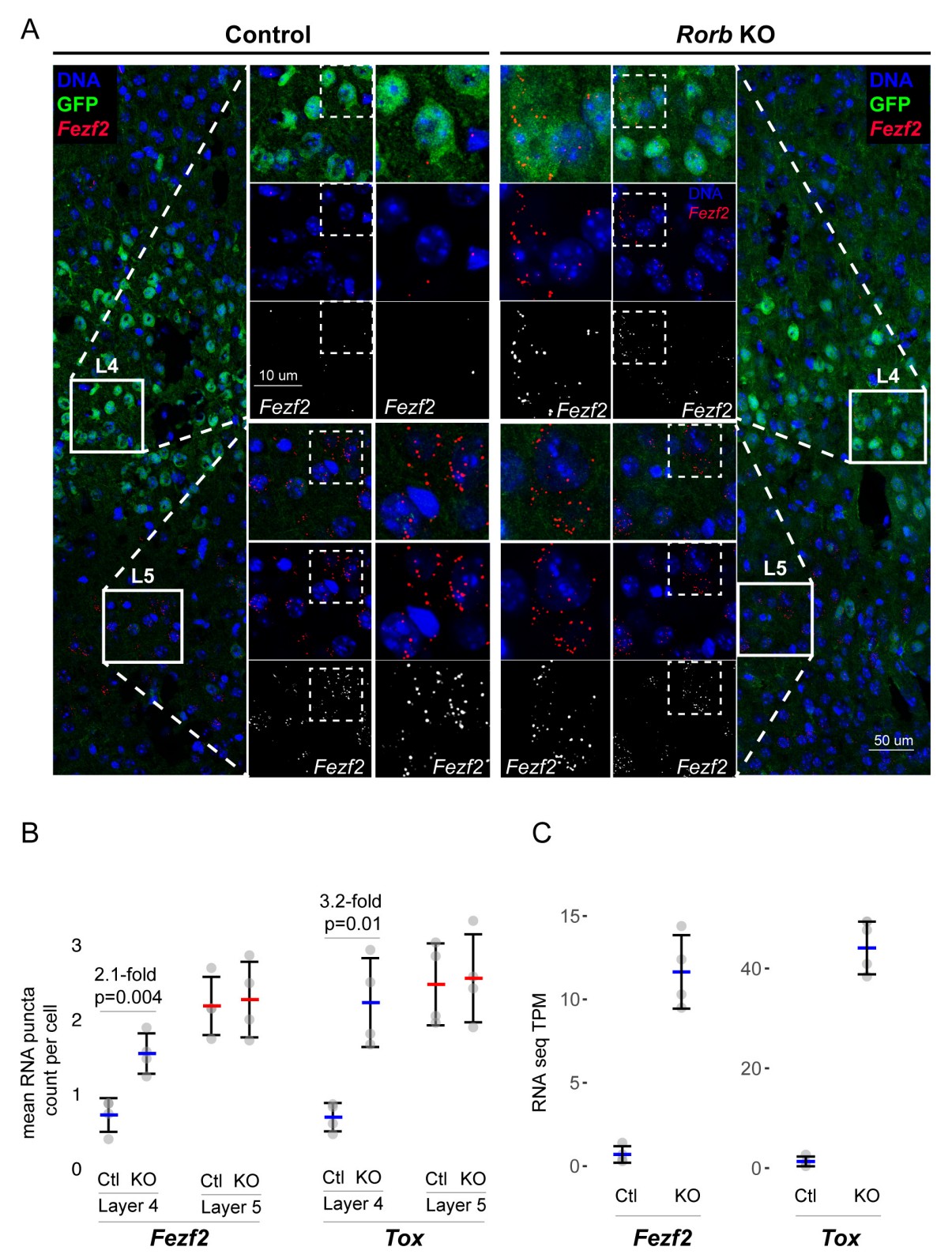

**Figure 5.** Confirmation of upregulated L5 genes in L4 neurons of *Rorb* KO. (**A**) RNAscope in situ hybridization of *Fezf2* in control (Ctl) and *Rorb* KO tissue. (**B**) Quantification of *Fezf2* and *Tox* RNA puncta per cell in either layer 4 or layer 5 of control and *Rorb* KO tissue. *Tox* images are shown in a *Figure 5—figure supplement 1*. N = 4 P30 animals for each genotype (Ctl or KO). Two regions containing S1 were averaged per animal. Whisker plots show the median per animal ± standard deviation. Gray points show mean number of puncta per cell for each layer in each animal. P-value by

*Figure 5 continued on next page*

*Figure 5 continued*

independent sample t-test. (C) RNA-seq changes in *Fezf2* and *Tox* expression at P30 replotted from heatmap of *Figure 4*. Gray points show values for individual replicates. Whisker plots show the mean ± standard deviation (N = 4).

The online version of this article includes the following figure supplement(s) for figure 5:

**Figure supplement 1.** RNAscope in situ hybridization of *Tox* in control (Ctl) and *Rorb* KO tissue.

expression. Foxo1 motifs were enriched near genes down-regulated at P2 and P7. Consistent with a role in early gene regulation, Foxo1 was highly expressed at P2 and declined with age in control neurons (*Figure 4D*). However, in the KO, Foxo1 remained significantly elevated at P7 eventually decreasing to levels comparable to control at P30. The close proximity of Foxo1 binding sites to down-regulated genes and its elevated expression at younger ages suggests it may act as a repressor that is normally silenced just after barrel formation to allow proper gene induction in L4 neurons. Without RORβ, silencing of Foxo1 is delayed allowing it to aberrantly repress targets at younger ages.

Interestingly, we did not find RORβ motifs enriched near L4 genes suggesting the shift in layer-specific gene expression is a downstream effect of RORβ loss. While RORβ does not appear to directly regulate layer-specific genes, Zfp281 motifs were enriched near L4 genes in the de novo motif search and confirmed by specific mapping (*Figure 6—figure supplement 1C* and *Figure 6D*). Zfp281 was highly expressed in both samples, at all ages, and unchanged by *Rorb* KO (*Figure 6—figure supplement 1C*). Zfp281 motifs were also enriched in regions that closed in the *Rorb* KO suggesting it might be a novel activator of L4-specific genes and dependent on some other factor to maintain accessible chromatin at its binding sites.

Nfe2l and NeuroD1 motifs were enriched near L5 genes. NeuroD1 motifs were also enriched in regions that closed suggesting it might act as an inhibitor of L5-specific genes as these genes increased expression when NeuroD1 sites closed. Nfe2l consists of a family of TFs that share a binding motif. Nfe2l1 was expressed at younger ages and increased in the adult while Nfe2l3 was highly expressed at P2 and silenced by P7 (*Figure 6—figure supplement 1D*). *Rorb* KO did not significantly disrupt expression of either, but the motif was enriched in regions that opened suggesting Nfe2l1 and/or three may be novel activators of L5-specific genes.

The TF motifs enriched near up-regulated DEGs were noteworthy for possible relationships with neuronal activity. Nr4a1 is an activity induced TF that regulates the density and distribution of excitatory synapses (*Chen et al., 2014*). Nfil3 and Hlf bind and compete for similar DNA motifs (*Mitsui et al., 2001*), and may also be involved in activity-regulated transcription. Nfil3 is up-regulated in human brain tissue following seizures (*Beaumont et al., 2012*), and mutations in Hlf are linked to spontaneous seizures (*Gachon et al., 2004*; *Hawkins and Kearney, 2016*). In addition, motifs for the classic immediate early genes, Jun and Fos, were enriched in regions that opened. These observations led us to examine the expression of other activity-regulated TFs. Many were significantly up-regulated at P30 while Lmo4 and its binding partner Lbd2 were up-regulated at P7 (*Figure 6—figure supplement 1E*; *Matsui et al., 2013*). Lmo4 expression is induced by calcium signaling and is required for TCA patterning in barrel cortex (*Kashani et al., 2006*; *Huang et al., 2009*). Another activity-regulated TF, Btbd3, which drives L4 neurons to orient their dendrites into barrel hollows, was significantly down-regulated (*Figure 6—figure supplement 1E*). Lmo4 and Btbd3 are the only genes previously shown to disrupt barrels that were also dysregulated in the *Rorb* KO (*Figure 6—figure supplement 1F*). In the *Rorb* KO *Lmo4* was up-regulated, but *Lmo4* KO disrupts barrels, suggesting that *Rorb* KO disrupts barrels through a divergent mechanism from what has been previously described.

Interestingly, the protein product of *S100A10*, p11, is involved in serotonin signaling via binding to the serotonin receptors Htr1b, Htr1d, and Htr4 (*Warner-Schmidt et al., 2009*). *S100A10* was down-regulated at P7 and P30 (*Figure 6—figure supplement 1G*). Htr1b was the only of the three serotonin receptor subunits known to interact with p11 expressed in our samples and was also significantly down-regulated at P7 and P30. These data suggest that in addition to altered layer identity, *Rorb* KO may also disrupt serotonergic signaling, an important pathway in TCA communication with cortex (*Kawasaki, 2015*). Together with up-regulation of activity-regulated TFs, L4 neurons in the *Rorb* KO likely have significantly altered responses to activity.

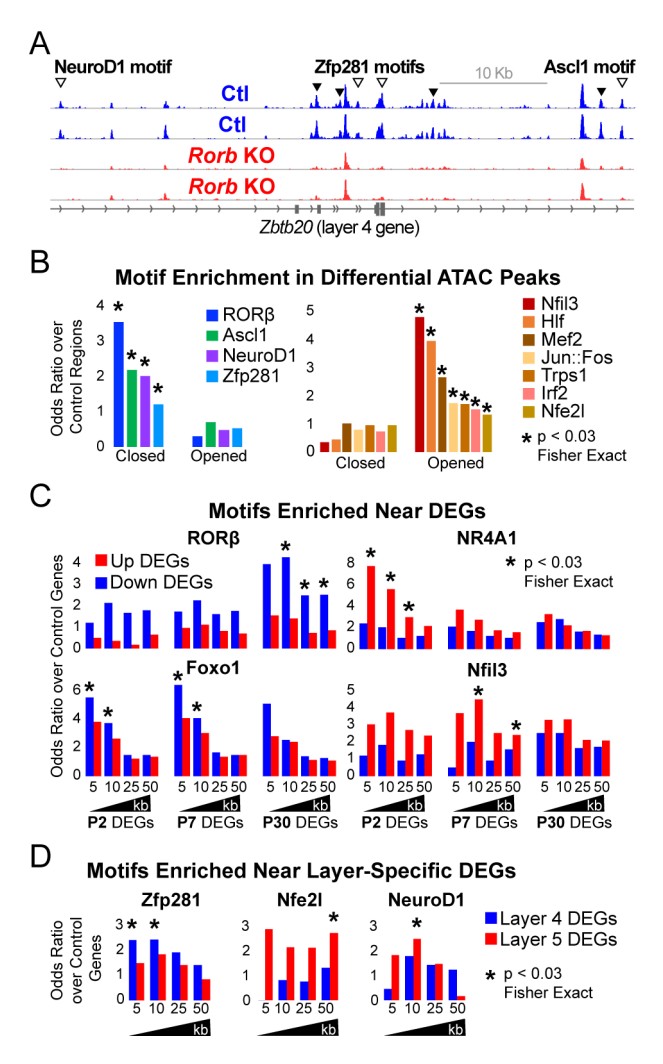

**Figure 6.** *Rorb* KO disrupts transcription factor binding sites near DEGs. (A) ATAC-seq normalized reads per million (RPM) for biological replicates, y-axis scaled 0–2. Samples collected from GFP⁺ S1 L4 *Rorb*^GFP/+ neurons (Ctl, blue) and GFP⁺ S1 L4 *Rorb*^GFP/GFP neurons (KO, red). Arrows indicate differential peaks (fold change ≥2, FDR < 0.02). Open arrows indicate differential peaks with transcription factor motif sequences as in (B). (B) Cross-validated motifs with significant enrichment in ATAC peaks with differential accessibility. Closed; regions with significantly reduced access, Opened; regions with significantly increased access in the *Rorb* KO. Motif instances were cross-validated between MEME and HOMER algorithms. Odds ratio and p-value calculated comparing to motif frequency in control regions. (C–D) Cross-validated motif enrichment in ATAC peaks near the TSSs of (C) up-regulated or down-regulated DEGs and (D) L4- or 5-specific genes. Bars plot odds ratio over control regions. Asterisk indicates significant motif enrichment (p<0.03 by Fisher exact test) in nearby ATAC peaks compared to control regions and separately significant enrichment (p<0.03 by Fisher exact test) of DEGs with a nearby motif compared to an independent group of control genes.

The online version of this article includes the following figure supplement(s) for figure 6:

**Figure supplement 1.** *Rorb* KO disrupts transcription factor binding sites near DEGs.

These analyses paint a complex picture where gene expression in L4 *Rorb* KO neurons is disrupted by multiple mechanisms. Loss of RORβ results in closure of many RORβ binding sites which are also enriched near genes with reduced expression in adults consistent with an activator role for RORβ. Other regulatory changes involve complex combinations of altered TF expression and/or altered binding potential at sites that opened or closed in the KO likely due to downstream effects

of RORβ loss. These changes impact both known neurodevelopmental regulators as well as activity-regulated TFs.

## *Rorb* KO delays excitatory input to barrel cortex

To examine whether RORβ loss impacts network activity, we examined inhibitory and excitatory synaptic properties of L4 neurons. We found no change in inhibitory innervation at P14 or P24 as measured by miniature inhibitory postsynaptic currents (mIPSCs), *Figure 7—figure supplement 1A-B*. However, synaptic function as measured by miniature excitatory postsynaptic currents (mEPSCs) revealed a significant delay in excitatory input, *Figure 7A–C*. At P5, the frequency of mEPSCs was low and comparable in control and KO, *Figure 7B–C*. At P7, around the time when recurrent cortical synapses begin to sharply increase (*Ashby and Isaac, 2011*) and LTP has just ended (*Crair and Malenka, 1995*), controls showed increased mEPSC frequency. However, *Rorb* KO animals had a significantly lower mEPSC frequency at P7 (*Figure 7A–C*), suggesting decreased functional synaptic input. At P10, *Rorb* KO neurons increased mEPSC frequency to levels comparable with controls. This suggests synaptic connections were delayed by *Rorb* KO mostly likely affecting recurrent excitatory connections. At P10, this defect in frequency is mostly corrected, but *Rorb* KO also showed significantly increased mEPSC amplitude at P10, possibly compensating for the delay at P7. By P19, both frequency and amplitude of mEPSCs were similar between control and KO (*Figure 7B*). These data support a subtle functional disruption to the barrel circuit in *Rorb* KO animals that is consistent with the transcriptional changes.

## The putative RORβ target, *Thsd7a*, is required for adult TCA, but not barrel wall organization

To begin exploring the relationship between disrupted gene expression in the *Rorb* KO and barrel organization, we examined known functions of genes differentially expressed at multiple developmental time points. Two candidates were identified with potential roles in cell migration and synaptogenesis. PlexinD1 (Plxnd1) is a cell signaling molecule known to play a role in pathfinding and synaptogenesis (*Chauvet et al., 2007*; *Wang et al., 2015*). Thrombospondin 7a (Thsd7a) regulates endothelial cell migration (*Wang et al., 2010*), but its role in the brain is unknown. In controls, expression of both genes followed a similar developmental trajectory as *Rorb*, peaking around P7 (*Figure 8A*). In the *Rorb* KO, *Plxnd1* was significantly lower at P2 and P7 while *Thsd7a* was significantly lower at all three time points. In addition, we identified several differential ATAC peaks near *Thsd7a* with

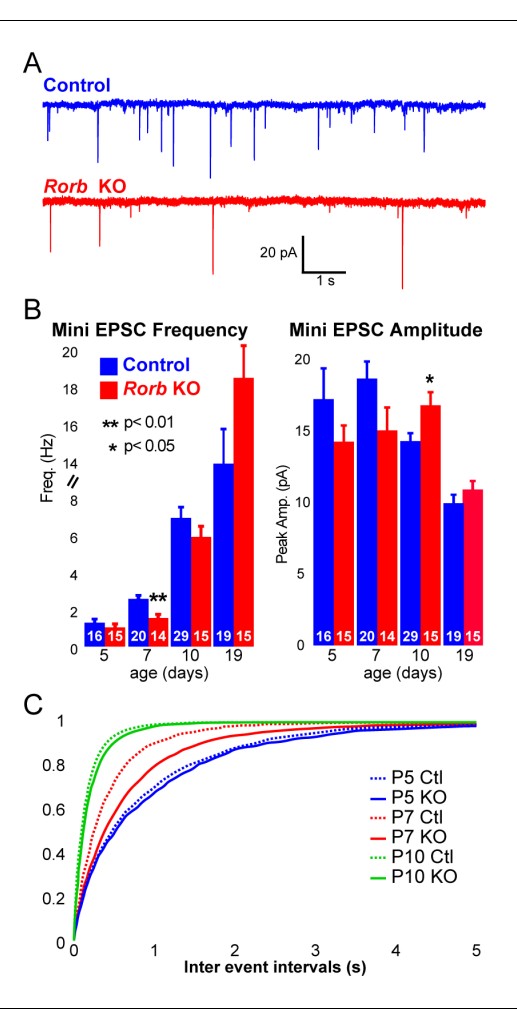

**Figure 7.** *Rorb* KO delays excitatory input to barrel cortex. (**A**) Example of miniature excitatory postsynaptic currents (mEPSCs) from L4 barrel cortex at P7. (**B**) Average mEPSC frequency and amplitude from Ctl and *Rorb* KO L4 barrel cortex at P5, P7, P10 and P19. Bars plot mean + SE, number of cells in parentheses. P values by 2-way ANOVA adjusted for multiple comparisons. (**C**) Cumulative histogram of inter-event intervals for control and *Rorb* KO L4 barrel cortex at P5, P7, and P10.

The online version of this article includes the following figure supplement(s) for figure 7:

**Figure supplement 1.** *Rorb* KO has minor effects on inhibitory input.

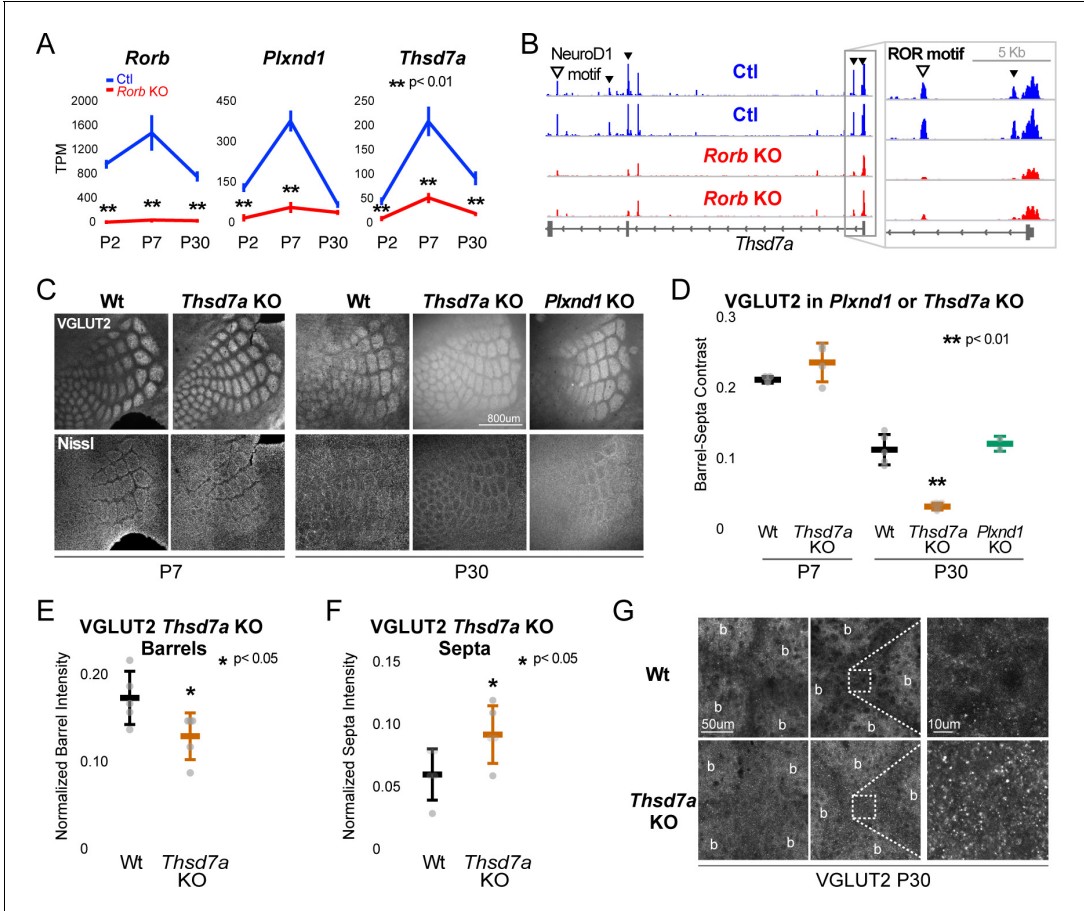

**Figure 8.** Thsd7a is required for TCA but not barrel wall organization. (**A**) Line plots of transcripts per million (TPM) measured by RNA-seq for three genes (*Rorb, Thsd7a,* and *Plxnd1*) from Ctl (blue) or *Rorb* KO (red) S1 layer IV barrel cortex. Lines plot the mean ± SE. (**B**) ATAC-seq around the *Thsd7a* gene (as in *Figure 6A*), y-axis scaled 0–3. (**C**) VGLUT2 and Nissl staining of barrel cortex at P7 and P30 from wild-type (Wt), *Plxnd1* KO, or *Thsd7a* KO. (**D**) Quantification of VGLUT2 Barrel-Septa Contrast from genetic lines in C. N = 2–5 animals. Whisker plots as described for *Figure 1B*. Statistical analysis summarized in *Figure 8—figure supplement 1A*. (**E**) Background normalized quantification of VGLUT2 contrast in barrel hollows. Two tissue sections containing the largest portions of whisker barrel field were averaged per animal. N = 5, P30 animals per genotype. Whisker plots as described for *Figure 1B*. (**F**) Background normalized quantification of VGLUT2 contrast in septa. Two tissue sections containing the largest portions of whisker barrel field were averaged per animal. N = 5, P30 animals per genotype. Whisker plots as described for *Figure 1B*. (**G**) VGLUT2 staining imaged at high magnification (63X) in P30 Wt or *Thsd7a* KO whisker barrel cortex. Barrels are labeled 'b'.

The online version of this article includes the following figure supplement(s) for figure 8:

**Figure supplement 1.** Barrel-septa contrast of VPM-specific afferents.

reduced accessibility (*Figure 8B*). This included a peak containing a strong RORβ motif just downstream of the transcription start site, suggesting *Thsd7a* might be a direct target of RORβ regulation.

There was no detectable disruption to barrel organization in *Plxnd1* conditional KO mice (*Plxin-D1flox* crossed to *Emx1cre*, *Figure 8C–D*). A *Thsd7a* constitutive KO also showed no disruption to barrel wall organization at P7 or P30. Interestingly, *Thsd7a* KO did show decreased VGLUT2 contrast between barrels and septa at P30 but not P7, suggesting Thsd7a is important for maintenance of TCA organization in adulthood (*Figure 8C–D*). The barrel phenotype of *Thsd7a* KO was qualitatively different from *Rorb* KO barrels. Specifically, the overall barrel pattern remained more intact in the *Thsd7a KO* despite the quantitative decrease in VGLUT2 contrast. As before, desegregation of VPM afferents was confirmed by VPM injection of AAV-hSyn-mCherry (*Figure 8—figure supplement 1*). *Thsd7a* KO may maintain sharper barrel borders than *Rorb* KO due to intact barrel walls. Reduction in VGLUT2 contrast in the *Thsd7a* KO could be due to increased TCA localization in the septa and/or decreased TCA localization in the barrels. To distinguish these two possibilities, three regions of

low VGLUT2 staining adjacent to the barrel field were quantified and used for within tissue slice normalization of barrel and septa intensities. *Thsd7a* KO resulted in a 24% decrease in barrel hollow VGLUT2 signal and a 56% increase in the septa (*Figure 8E–F*). High resolution imaging showed a clear increase in VGLUT2 puncta located in the septa (*Figure 8G*). Thus, loss of Thsd7a after *Rorb* KO likely contributes to the decrease in TCA segregation in adulthood.

## Discussion

While somatotopic maps were one of the earliest and most obvious forms of cytoarchitecture, our understanding of the role neuronal identity plays in module formation is largely unknown. Studies have long approached the question of what drives cortical organization from the perspective of network activity and, in the case of barrel cortex, from the perspective of key structures and pathways needed to relay sensory input. More recent studies characterizing transcription factors required for barrel organization point to the importance of molecular mechanisms regulating transcriptional programs. However, many of these TFs are part of the pathways that carry sensory input or are fundamental regulators of broad developmental programs. It was unclear whether a TF such as RORβ, a highly restricted marker of L4 identity, could influence macro-scale processes such as module formation. Indeed, we show that while RORβ is clearly regulating only a fraction of the phenotypic and transcriptional properties of L4 neurons, it is necessary for terminal specification of L4 identity and proper organization of L4 cytoarchitecture.

Specifically, RORβ is required in the cortex for barrel wall formation and full TCA segregation. This differs from earlier work focusing on the role of TCA patterning and activity as instructive for barrel wall formation. Instead, we find that loss of RORβ specifically in the cortex affects TCA segregation shortly after barrel walls should have formed, suggesting that bidirectional signaling between L4 neurons and TCAs is involved in establishing proper organization. That such signaling occurs was first suggested by cortex-specific knockout of NMDA receptor subunits (*Iwasato et al., 2000*; *Lee et al., 2005*). A second study highlighting this role of cortical influence on TCA organization knocked out the metabotropic glutamate receptor *Grm5* (*Ballester-Rosado et al., 2016*) in cortical neurons. In contrast, cortex-specific knockout of another member of the ROR family of transcription factors, *Rora* (*Vitalis et al., 2018*) disrupts the cellular organization of cortical barrels, but appears to leave TCA segregation intact.

While loss of RORβ function affected TCA segregation from the time of formation we note that loss of the putative RORβ gene target, *Thsd7a*, primarily affected TCA segregation in adults despite maximal expression at P7. While additional studies are needed, we speculate one possible explanation could be that Thsd7a functions around the time of barrel formation to establish long lasting TCA structures that only manifest aberrant phenotypes later in life. Alternatively, the moderate expression level of Thsd7a at P30 may be sufficient for a role in adult maintenance. In either case, a role for Thsd7a in the nervous system has not been described previously. In endothelial cells, Thsd7a localizes to the membrane of the leading edge of migrating cells where it functions to slow or inhibit migration (*Wang et al., 2010*). Perhaps in somatosensory cortex it inhibits movement of nearby projections such as dendrites or axons allowing cortical neurons to 'corral' TCAs in barrel hollows. *Thsd7a* is not the only potential downstream target of RORβ worthy of further investigation. Pcdh20 has a role in L4 identity in regulating appropriate laminar positioning of L4 cells. Without Pcdh20, cells migrate to L2/3 instead (*Oishi et al., 2016*). In RORβ KO cells, *Pcdh20* is down-regulated but cells still migrate to L4 suggesting a possible novel role for Pcdh20 downstream of RORβ function.

Our observation that barrel organization declined with age in wildtype animals is very interesting and possibly the first description of this phenomenon in mice (*Rice, 1985*). It suggests continued plasticity or degradation of maintenance mechanisms over time. Few studies have examined plasticity within this structure in adulthood. This is in part because studies have shown a decline in the capacity to rewire sensory input to the cerebral cortex with age in certain systems. In the visual system, loss of sensory input has been shown to alter TCAs during a critical postnatal period (*Antonini and Stryker, 1993*; *Erzurumlu and Gaspar, 2012*). It is thought that once this critical period closes, TCA organization is fixed. Thus, developmental processes in the visual and somatosensory systems are assumed to stabilize TCAs and restrict learning and memory related changes to plasticity among cortical connections (*Fox, 2002*; *Feldman and Brecht, 2005*; *De Paola et al., 2006*; *Karmarkar and Dan, 2006*). However, there is some evidence to support a shift in this model

of adult plasticity in both the visual and somatosensory cortex (*Khibnik et al., 2010*; *Wimmer et al., 2010*). In particular, Oberlaender et al. showed that a mild form of sensory deprivation induced by whisker trimming in 3 month old rats substantially altered TCAs in barrel cortex (*Oberlaender et al., 2012*). However, because adult TCA plasticity has garnered limited attention, we currently lack genetic studies examining the molecular mechanisms behind these processes. The natural decline in barrel organization and the mechanism of Thsd7a influence on TCA segregation merit further investigation as exciting new contexts to study both the functional roles of cortical organization and the impact of age.

Recent studies are revealing that neuronal identity in certain structures remains plastic during early postnatal periods. For example, mistargeted L4 neurons that migrate to layer 2/3 take on characteristics of their surroundings (*Oishi et al., 2016*) and misexpression of some TF can alter the identity of postnatal neurons (*Rouaux and Arlotta, 2010*; *Rouaux and Arlotta, 2013*). We find that loss of RORβ disrupts the transcriptional identity of L4 neurons, which lose expression of many L4 genes and aberrantly express many L5 genes. While this shift to a more L5-like transcriptional profile is not a global identity switch, it suggests L4 identity continues to be refined relative to deeper layer profiles late into postnatal development.

The complex expression changes observed likely occur through a multi-tiered reorchestration of gene regulation. Up-regulation of known L5 TFs such as Bcl11b/Ctip and Etv1 at P2 may help drive an early diversion down an L5-like trajectory. Regulatory signatures detected in adult neurons such as closure of binding sites for Zfp281 enriched near L4 genes and opening of Nfe2l1/3 motifs enriched near L5 genes may represent the tip of the developmental iceberg. In addition, our stringent motif analysis aimed to keep false positives low may also miss relevant regulators with more minor roles. While we detect changes in binding capacity for many TFs, including RORβ, the complexity of dysregulation spread out across early postnatal development means there are certainly additional mechanisms driving this shift in cellular identity to be discovered. Here we combine the power of genetic knock-out strategies with multiple molecular profiling assays to interrogate the transcriptional network influenced by RORβ. We found RNA-seq paired with ATAC-seq provided a rich picture of the transcriptional changes occurring in *Rorb* KO neurons and insight into both developmental and adult functioning. Changes to the transcriptional network involved both differentially expressed TFs and TFs whose only perturbation was increased or decreased access to binding sites. Without these complementary perspectives, proteins such as Zfp281 and Nfe2l1/3 TFs might have been overlooked.

We identify several other TFs worthy of further investigation for their role in cortical development. Ascl1 and NeuroD1 are potent TFs that can induce transdifferentiation of mouse embryonic perinatal fibroblasts or microglia into neurons (*Vierbuchen et al., 2010*; *Matsuda et al., 2019*). NeuroD1 binds a different motif than NeuroD2, which is known to regulate barrel formation (*Ince-Dunn et al., 2006*), suggesting a distinct role. In addition, Trps1 was strongly up-regulated by RORβ loss at P7 and P30, and it was enriched in regions that opened. Its role in neurons is not clear, but it has been characterized as a transcriptional repressor that inhibits cell migration making it a tempting target to explore the lack of L4 neuron migration necessary to form barrel walls (*Wang et al., 2018*).

In addition to disrupted layer identity, we also detect a significant disruption in the potential for *Rorb* KO cells to transcriptionally respond to activity connecting cellular identity, module formation and molecular responsiveness to input. In the adult *Rorb* KO, many activity-regulated TFs were up-regulated, with the exception of Btbd3, and their DNA motifs showed increased accessibility. Around P7, when activity is critical for instructing cortical reorganization, we see reduced mEPSC frequency in L4 *Rorb* KO neurons, which is rectified by P10. Some of the transcriptional changes in the *Rorb* KO may be a form of compensation for the lack of input at P7. Failed up-regulation of Htr1b and down-regulation of *S100a10*/p11 may also be an attempt to increase activity in KO neurons. More is known about the role of Htr1b in TCAs where it is transiently expressed and, when stimulated, inhibits thalamic neuronal firing (*Bennett-Clarke et al., 1993*; *Rhoades et al., 1994*) and disrupts barrel formation (*Young-Davies et al., 2000*). TCA inhibition is thought to be the mechanism by which excess 5-HT disrupts barrels. While it is difficult to infer the role of Htr1b and p11 without characterizing cellular localization in S1 L4 neurons, down-regulation of p11 resulting in less Htr1b localizing to the membrane coupled with reduced Htr1b expression could relieve inhibition in L4 *Rorb* KO neurons. Barrel formation and the ability to respond to activity inputs corresponds with increased RORβ expression and this increase is attenuated when TCA inputs are eliminated

(*Pouchelon et al., 2014*). Together this suggests terminal differentiation and migration of neurons within L4 to form barrel walls are closely synchronized to excitatory input and require RORβ for proper establishment.

Although few other studies have examined the transcriptional targets and molecular mechanisms of TFs that regulate barrel formation, our study suggests RORβ is likely involved in the later stages of cellular specification and implicates several new TFs. RORβ also appears to function by distinct mechanisms from TFs previously characterized to regulate barrel formation. Loss of Bhlhe22 disrupts both barrel wall formation and TCA segregation but results in down-regulation of Lmo4 (*Joshi et al., 2008*) unlike *Rorb* KO, which increased Lmo4. Interestingly, Eomes is required for barrel wall organization but does not appear to affect TCA segregation (*Elsen et al., 2013*). Lhx2 and RORα are more broadly expressed than RORβ. *Lhx2* KO results in moderate down regulation of RORβ suggesting it is also likely upstream of RORβ in barrel development (*Wang et al., 2017*). Loss of Lhx2 greatly reduced TCA branching producing smaller barrels and barrel field. This phenotype is very similar to *Rora* KO barrels (*Vitalis et al., 2018*) suggesting RORα's mechanism may be more similar to earlier developmental TFs than to RORβ. Disruption of barreloid development in *Rora* KO thalamus is also consistent with a role in earlier stages of development (*Vitalis et al., 2018*). However, *Rora* was down-regulated in our *Rorb* KO dataset suggesting it may also have a role downstream of RORβ. Several additional TFs appear to be downstream of RORβ. For example, Btbd3 is important for dendritic orientation and is down-regulated in the *Rorb* KO. It may be that dendritic orientation occurs after L4 cells have migrated to form barrel walls and provide an organized reference point for orientation. Thus, we have characterized in depth the molecular and transcriptional mechanism of RORβ as it orchestrates a critical juncture in barrel development where terminal differentiation and activity inputs are integrated to drive cellular organization in the cortex.

## Materials and methods

### Key resources table

| Reagent type (species) or resource | Designation | Source or reference | Identifiers | Additional information |
|---|---|---|---|---|
| Genetic reagent (*M. musculus*) | *Rorb$^{GFP}$* (*Rorb$^{1g}$*) | PMID:23652001 | | Dr. Douglas Forrest (Laboratory of Endocrinology and Receptor Biology, National Institutes of Health) |
| Genetic reagent (*M. musculus*) | *Rorb$^{f/f}$* (*Rorb$^{flox/flox}$*) | PMID:29224725 | | Dr. Douglas Forrest (Laboratory of Endocrinology and Receptor Biology, National Institutes of Health) |
| Genetic reagent (*M. musculus*) | *Rosa26$^{tdTomato}$* | Jackson Laboratories | RRID:IMSR_JAX:007909 | |
| Genetic reagent (*M. musculus*) | *plexinD1$^{flox}$* | Jackson Laboratories | RRID:IMSR_JAX:018319 | |
| Genetic reagent (*M. musculus*) | *Thsd7a KO* | Jackson Laboratories | RRID:MGI:6263683 | |
| Genetic reagent (*M. musculus*) | *Emx1$^{cre}$* | Jackson Laboratories | RRID:IMSR_JAX:005628 | |
| Genetic reagent (*M. musculus*) | *Sert$^{Cre}$* | Jackson Laboratories | RRID:IMSR_JAX:014554 | |
| Genetic reagent (*M. musculus*) | *CamK2a$^{cre}$* | Jackson Laboratories | RRID:IMSR_JAX:005359 | |
| Antibody | Guinea pig anti-VGLUT2 | Millipore AB2251 | RRID:AB_2665454 | 1:500-1:1000 |

*Continued on next page*

*Continued*

| Reagent type (species) or resource | Designation | Source or reference | Identifiers | Additional information |
|---|---|---|---|---|
| Antibody | rabbit anti-VGLUT2 | Synaptic Systems 135 403 | RRID:AB_887883 | 1:250 |
| Antibody | chicken anti-GFP | Aves labs GFP-1020 | RRID:AB_10000240 | 1:500-1:1000 |
| Antibody | Goat Anti-Rabbit Alexa Fluor 564 | Invitrogen A-11037 | RRID:AB_2534095 | 1:500 |
| Antibody | Goat Anti-Chicken Alexa Fluor 488 | Invitrogen A-11039 | RRID:AB_2534096 | 1:500 |
| Antibody | Goat Anti-Rabbit Alexa Fluor 633 | Invitrogen A-21070 | RRID:AB_2535731 | 1:500 |
| Antibody | Goat Anti-Guinea Pig Alexa Fluor 647 | Invitrogen A-21450 | RRID:AB_2735091 | 1:500 |
| Stain | Nissl | Invitrogen N21479 | | 1:250 |
| Other | AAV-hSyn-mCherry | Addgene 114472-AAV8 | RRID:Addgene_114472 | Undiluted |
| Commercial assay or kit | RNAscope Fluorescent Multiplex kit | Advanced Cell Diagnostics, 320850 | | |

## Animals

All animals were bred, housed, and cared for in Foster Biomedical Research Laboratory at Brandeis University (Waltham, MA, USA). Animals were provided with food and water ad libitum and kept on a 12 hr:12 hr light:dark cycle. Cages were enriched with huts, chew sticks, and tubes. All experiments were approved by the Institutional Animal Care and Use Committee of Brandeis University, Waltham, MA, USA.

*Rorb$^{GFP}$* (*Rorb$^{1g}$*) and *Rorb$^{f/f}$* (*Rorb$^{flox/flox}$*) mice were obtained from Dr. Douglas Forrest (*Liu et al., 2013*; *Koch et al., 2017*; *Byun et al., 2019*). *Rorb$^{GFP}$* mutation deletes the RORβ1 isoform, the predominant isoform in brain, and not the RORβ2 isoform (*Liu et al., 2013*). The *Rorb$^{f/f}$* allele deletes both isoforms. The following mice were obtained from Jackson Laboratories: *Rosa26$^{td-Tomato}$* (stock 007909, RRID:IMSR_JAX:007909); *plexinD1$^{flox}$* (stock 018319, RRID:IMSR_JAX:018319); *Thrombospondin7a* KO (*Thsd7a*) (stock 027218, RRID:MGI:6263683); *Emx1$^{cre}$* (stock 005628, RRID: IMSR_JAX:005628); *Sert$^{Cre}$* (Slc6a4) (stock 014554, RRID:IMSR_JAX:014554). *CamK2a$^{cre}$* (stock 005359, RRID:IMSR_JAX:005359).

## Perfusion

Animals were fatally anesthetized and transcardially perfused with 15 mL 1x PBS (Fisher, SH3001304) then 15 mL 4% PFA (Sigma Aldrich P6148-500G). Brains were fixed overnight in tangential orientation. After removing the whole brain from the skull, the cerebellum and olfactory bulbs were removed. The brain was split into two hemispheres along the longitudinal fissure and the midbrain was gently excised. The remaining cortex was placed in a shallow well made from a cryostat mold, filled with 4% PFA and a glass slide set on top for flattening. Brains were removed from PFA after 24–48 hr and stored in 30% sucrose/PBS solution at 4˚C.

## Immunohistochemistry

50 μm slices were made on a freezing Microtome (Leica SM 2010R). Controls and KOs were stained together in batches. Slices were permeabilized overnight at 4˚C in 0.3% Triton-X100 (Sigma Aldrich, T8787) and 3% Bovine Serum Albumin (Sigma B4287-25G) in PBS. Slices were then incubated for 24 hr in primary antibody solution containing 0.3% Triton-X100% and 3% Bovine Serum Albumin (BSA) in PBS at 4˚C. Primary antibody dilutions were as follows: Guinea pig anti-VGLUT2 (Millipore AB2251, RRID:AB_2665454) 1:500-1:1000, rabbit anti-VGLUT2 (Synaptic Systems 135 403, RRID:AB_887883) 1:250, chicken anti-GFP (Aves labs GFP-1020, RRID:AB_10000240) 1:500-1:1000. Slices

were washed three times in PBS for 10 min each at room temp and then moved to secondary antibody solution containing 0.3% Triton-X100, 3% Bovine Serum Albumin, 10% normal goat serum. All secondaries were used at 1:500; Goat Anti-Rabbit Alexa Fluor 564 (Invitrogen A-11037, RRID:AB_2534095), Goat Anti-Chicken Alexa Fluor 488 (Invitrogen A-11039, RRID:AB_2534096), Goat Anti-Rabbit Alexa Fluor 633 (Invitrogen A-21070, RRID:AB_2535731), Goat Anti-Guinea Pig Alexa Fluor 647 (Invitrogen A-21450, RRID:AB_2735091). Slices were stained using Nissl (Invitrogen N21479) at 1:250 in PBS for 2 hr at room temperature, washed in PBS as before, and mounted in VECTASHIELD HardSet Mounting Medium (Vector Laboratories, H-1500, RRID:AB_2336787). Slides were stored at −20C and imaged within 1 week.

## Imaging and fluorescence quantification

Tissue was imaged on a Leica DMI 6000B Inverted Widefield Imaging Fluorescence Microscope or a Zeiss LSM 880 confocal microscope. All genotypes and age groups contained roughly even numbers of males and females. A minimum of two slices containing at least five intact barrels between rows B-D were quantified per animal. Experimenters were blinded to age and genotype during imaging and quantification. Regions of interest (ROIs) were drawn manually by a blinded researcher around 5–6 intact barrels from rows B, C, or D using Fiji (*Schindelin et al., 2012*). An ROI including the total space around selected barrels up to the edges of adjacent barrels was drawn to be used for calculating septa intensity (*Figure 9*). For *Thsd7a* KO and controls, three additional ROIs were drawn in the region adjacent to barrel cortex with low VGLUT2 signal to be used as background to normalize barrel and septa intensity. Custom MATLAB code was used to quantify the average fluorescence in ROIs. Septa intensity was calculated as septa total ROI intensity - sum(barrel ROIs). Contrast = (barrel - septa) / (barrel + septa). For absolute barrel or septa intensity, measurements were normalized to background regions (*Figure 9*) within each tissue section. This was not necessary for contrast calculations because contrast is a ratio. Contrast and normalized barrel and septa intensity were averaged for two slices per animal. Two-way ANOVA was used to test for a significant effect of genotype and/or age as well as for an interaction between the two variables. Independent sample t-test was used to test for significant differences between genotypes at each age. No power analysis was performed and numbers of replicates performed were the minimum needed to demonstrate reproducibility, consistent with practices in similar published studies.

## AAV injection into VPM

50 nl of AAV-hSyn-mCherry (Addgene 114472-AAV8, RRID:Addgene_114472) was delivered by stereotactic injection to the dorsal VPM of P18-20 animals. Mice were euthanized by cardiac perfusion of 4% paraformaldehyde solution at P30. Cortex was removed and flattened for tangential sectioning of barrel field into 50 µm slices on a freezing Microtome (Leica SM 2010R). Subcortical structures were embedded in agarose and sectioned into 50 µm coronal slices on a vibratome (Leica VT1000S), counterstained with DAPI, and imaged (Keyence BZ-X700). Barrel cortex was stained, imaged and contrast calculated as described above. We required two slices with a minimum of two mCherrry saturated barrels and no mCherry outside of the barrel field. Saturated barrels were defined as adjacent barrels surrounded by barrels with mCherry signal. Only saturated barrels were quantified and the same ROIs were used to quantify mCherry and VGLUT2.

## Multiplex fluorescent RNA in situ hybridization (RNAscope) with immunohistochemistry

Mice were euthanized by cardiac perfusion of 4% paraformaldehyde solution at P30. Brain tissue was pretreated according to the RNAscope Sample Preparation and Pretreatment Guide for Fresh Frozen Tissue (Manual RNAscope assay; Advanced Cell Diagnostics). Tissue was sectioned at 12 µm and subsequent staining

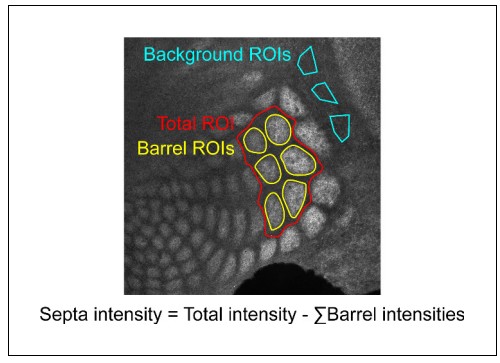

**Figure 9.** Example of quantification method. Regions of interest (ROIs) were drawn in Fiji by a researcher blinded to genotype and age.

performed according to the manufacturer's instruction for the RNAscope Fluorescent Multiplex kit (Advanced Cell Diagnostics, 320850) with two protocol modifications. Antigen retrieval was carried out in an autoclave set to a 5 min 'fast' cycle, 121°C, 15 psi. After protease III digestion, probe solutions containing 313301-C2 (*Fezf2*) or 484781 (*Tox*) also contained 10% NGS and 3% BSA to allow the probe binding step to also serve as the IHC blocking step. After developing the fluorescent in situ signal, slides were protected from light and stained overnight at room temperature with 1:250 chicken anti-GFP (Aves labs GFP-1020, RRID:AB_10000240) diluted in 1X Tris-borate-EDTA (TBE) buffer containing 10% NGS and 3% BSA. Slides were washed four times in 1X TBE for 2–5 min and incubated for two hours at room temperature with 1:500 Goat Anti-Chicken Alexa Fluor 488 (Invitrogen A-11039, RRID:AB_2534096). Slides were washed four times in 1X TBE for 2–5 min, counterstained with DAPI and coverslips mounted according to the instructions for the RNAscope Fluorescent Multiplex kit. Batches of staining were balanced to contain equal numbers of control and *Rorb* KO samples per batch.

Stained tissue was imaged on a Zeiss LSM 880 confocal microscope. Two regions of neocortex containing S1 were imaged for each animal with automated image stitching so that layers 2 through six were contained in a single image. Images of RNA signal were background subtracted in ImageJ (Fiji) using a rolling ball radius of 5 pixels. GFP signal was used to draw ROIs within L4 and L5. A custom CellProfiler (*Lamprecht et al., 2007*) pipeline identified cells by identifying nuclei from DAPI images and expanding ROIs, and identified RNA puncta. RNA puncta were associated with the nearest cell in R using X,Y coordinates output from CellProfiler. RNA puncta were tallied per cell and the mean calculated per image then per animal and plotted. P-values were calculated by independent sample t-test between Ctl and KO L4.

## Electrophysiology

*Rorb*$^{GFP/GFP}$ (KO) and *Rorb*$^{GFP/+}$ (control; Ctl) mice were anesthetized with isoflurane and decapitated. Coronal slices (300 μm) containing the primary somatosensory cortex were cut on a Leica (VT1000S) vibratome and incubated at room temperature in ACSF containing (mM) 126 NaCl, 25 NaHCO3, 2.5 KCl, 1.2 NaHPO4, 2 CaCl2, 1 MgCl2 and 32.6 dextrose adjusted to 326 mOsm, pH 7.4 and saturated with 95%/5% O2/CO2. Submerged, whole cell recordings were performed at 32 ± 1° on an upright microscope (Olympus BX50) equipped with epifluorescence. Pipettes with resistance 4–6 Mohm were filled with internal solution containing (mM) 100 K-gluconate, 20 KCl, 10 HEPES, 4 Mg-ATP, 0.3 Na-GTP, 10 Na-phosphocreatine and 0.2% biocytin adjusted to 300 mOsm, pH 7.35. For mIPSC recordings, the internal included 133 mM KCl and gluconate was omitted to bring $E_{Cl}$ to 0 mV. Recordings were made using an Axoclamp 700A amplifier, and were digitized at 10–20 kHz and analyzed using custom software running under Igor 6.03 (Wavemetrics). Miniature synaptic events were recorded in voltage clamp at −70 mV in the presence of PTX (mEPSCs) or DNQX+APV (mIPSCs) respectively.

Spiny stellate neurons were recognized based on their compact, GFP$^+$ cell bodies within the GFP$^+$ cell-dense layer 4. Input resistance was measured every 10–20 s with a small hyperpolarizing pulse and data were discarded if input or series resistance changed by >20%. P-values were calculated by 2-way ANOVA and adjusted for multiple comparisons by Tukey post hoc correction.

## RNA-seq

RNA-seq was performed as described previously (*Sugino et al., 2019*). Briefly, 1000–1500 GFP$^+$ cells were isolated by FACS (BD FACSAria Flow Cytometer) from micro dissected L4 S1 live tissue (N = 4 biological replicates per age and genotype). *Figure 10* shows examples of the region micro dissected out to exclude L5. The four independent biological samples were collected from a pool generated by combining tissue from one male and one female mouse for a total of 8 animals used per time point. Cells

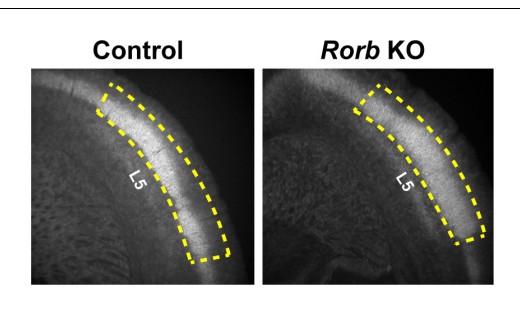

**Figure 10.** Example of micro dissected region for L4 S1 from coronal slices. Yellow dashed line indicates the tissue retained for FACS. Layer five is labeled for reference.

were sorted directly into extraction buffer and RNA stored at −80C for <three weeks. All libraries were prepared and sequenced in a single batch to prevent batch effects. Total RNA was purified (Arcturus PicoPure RNA Isolation kit, KIT0204) according to manufacturer's specifications. Libraries were prepared using Ovation Trio RNA-Seq library preparation kit with mouse rRNA depletion (0507–32) according to manufacturer's specifications and sequenced on a NextSeq Illumina platform (NextSeq 500/550 High Output (1 × 75 cycles)) obtaining 27 ± 2 million reads (mean ± SE). Reads were mapped by STAR with 90 ± 0.3% unique mapping (mean ± SE) and quantified with feature-Counts (*Liao et al., 2014*). Differentially expressed genes were identified by Limma (*Ritchie et al., 2015*) using a fold change cutoff of 2 and padj <0.01 from a moderated t-test adjusted for multiple comparisons using FDR (Benjamini-Hochberg).

## ATAC-seq

ATAC-seq was performed as described previously (*Clark et al., 2019*; *Sugino et al., 2019*). Briefly, 30,000–50,000 GFP$^+$ cells were isolated by FACS from microdissected L4 live tissue (N = 2 biological replicates per age and genotype). The two independent samples were collected from a pool generated by combining tissue from two male and two female mice for a total of 8 animals used. Nuclei were transposed for 30 min and libraries amplified according to published methods (*Corces et al., 2017*). Tagmented nuclei were stored at −20C for <two weeks. All ATAC libraries were purified, amplified, and sequenced as a single batch. Libraries were sequenced on a NextSeq Illumina platform (high output 300 cycles (2 × 150 bp)) producing 105 ± 24 (mean ± SE) million reads per replicate. Reads were mapped using Bowtie2 and filtered producing 24 ± 2 (mean ± SE) million unique non-mitochondrial reads per replicate. TSS enrichment calculated per replicate according to the ENCODE quality metric (*Corces et al., 2017*) (https://github.com/ENCODE-DCC/atac-seq-pipeline) was 34 ± 3 (mean ± SE). Peaks were identified permissively using HOMER (-style dnase –fdr 0.5 -minDist 150 -tbp 0 -size 75 -regionRes 0.75 -region) (*Heinz et al., 2010*) and IDR (threshold = 0.01, pooled_threshold = 0.01) was used to identify reproducible peaks (*Li et al., 2011*). Differential ATAC peaks were identified using DiffBind with an FDR threshold = 0.02 and log2 fold change in normalized read coverage threshold ≥1 (*Ross-Innes et al., 2012*).

## Data access

Raw and processed RNA-seq and ATAC-seq files are available at GEO accession GSE138001.

## Motif analysis

Motifs identified de novo from the sequences underlying ATAC peaks was carried out using MEME AME with shuffled input sequences as control and default settings (Fraction of maximum log-odds = 0.25, E-value threshold ≤10) (*McLeay and Bailey, 2010*), and HOMER findMotifsGenome.pl function masking repeats and -size given (*Heinz et al., 2010*). Scanning for specific motif matches in the sequences underlying ATAC peaks was carried out using MEME FIMO used the default threshold of p-value<1e-4 (*Grant et al., 2011*) and HOMER findMotifsGenome.pl -find function. When possible 2–3 PWMs were obtained from Jaspar (*Khan et al., 2018*) and Cis-BP (*Weirauch et al., 2014*) prioritizing PWMs from direct data sources such as ChIP-seq. The R package GenomicRanges (*Lawrence et al., 2013*) was used to identify overlapping motifs between the two algorithms for cross validation. The overlap criteria allowed a 1 bp difference in the start or end position of the motif to accommodate ambiguity among motif models. Fisher Exact tests were calculated in R to test for enrichment of motifs in ATAC regions compared to control regions and to test for enrichment of genes with a nearby motif from a DEG group compared to a control group of genes. The set of control regions was generated by shuffling ATAC peaks throughout the genome excluding sequence gaps using BedTools (*Quinlan and Hall, 2010*) and the control group of genes were defined as expressed above 5 TPM but unchanged by age or *Rorb* KO.

## Acknowledgements

We thank Dr. Roland Schüle for agreeing to share the *Rorb*$^{f/f}$ line, and Dr. Matthew Eaton for friendly bioinformatic advice. Supported in part by the intramural research program at NIDDK at the National Institutes of Health (DF).

## Additional information

### Competing interests

Sacha B Nelson: Reviewing editor, *eLife*. The other authors declare that no competing interests exist.

### Funding

| Funder | Grant reference number | Author |
|---|---|---|
| National Institute of Neurological Disorders and Stroke | NS109916 | Erin A Clark<br>Michael Rutlin<br>Lucia S Capano<br>Samuel Aviles<br>Jordan R Saadon<br>Praveen Taneja<br>Qiyu Zhang<br>James B Bullis<br>Timothy Lauer<br>Emma Myers<br>Anton Schulmann |

The funders had no role in study design, data collection and interpretation, or the decision to submit the work for publication.

### Author contributions

Erin A Clark, Conceptualization, Data curation, Formal analysis, Supervision, Validation, Investigation, Visualization, Methodology, Project administration; Michael Rutlin, Conceptualization, Resources, Formal analysis, Supervision, Validation, Investigation, Visualization, Project administration; Lucia S Capano, Data curation, Formal analysis, Supervision, Validation, Investigation, Visualization, Methodology; Samuel Aviles, Data curation, Software, Formal analysis, Investigation, Visualization, Methodology; Jordan R Saadon, Data curation, Validation, Investigation, Visualization, Methodology; Praveen Taneja, James B Bullis, Formal analysis, Investigation; Qiyu Zhang, Timothy Lauer, Investigation; Emma Myers, Data curation, Formal analysis; Anton Schulmann, Data curation; Douglas Forrest, Resources; Sacha B Nelson, Conceptualization, Resources, Supervision, Funding acquisition, Project administration

### Author ORCIDs

Erin A Clark  https://orcid.org/0000-0002-4013-325X
Lucia S Capano  http://orcid.org/0000-0003-3470-9360
Qiyu Zhang  http://orcid.org/0000-0002-7141-4046
Sacha B Nelson  https://orcid.org/0000-0002-0108-8599

### Ethics

Animal experimentation: All experiments were conducted in accordance with the requirements of the Institutional Animal Care and Use Committees at Brandeis University (protocol #17001).

### Decision letter and Author response

Decision letter https://doi.org/10.7554/eLife.52370.sa1
Author response https://doi.org/10.7554/eLife.52370.sa2

## Additional files

### Supplementary files
• Transparent reporting form

## Data availability

Raw and processed RNA-seq and ATAC-seq files are available at GEO accession GSE138001.

The following dataset was generated:

| Author(s) | Year | Dataset title | Dataset URL | Database and Identifier |
|---|---|---|---|---|
| Nelson SB, Clark EA, Myers E, Schulmann A | 2019 | Cortical RORb is required for layer 4 transcriptional identity and barrel integrity | http://www.ncbi.nlm.nih.gov/geo/query/acc.cgi?acc=GSE138001 | NCBI Gene Expression Omnibus, GSE138001 |

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
