## [Decision Letter]

**Acceptance summary:**

This study provides insights into the molecular mechanisms underlying RORβ's involvement in barrel formation and thus provides a tantalizing link across gene expression, chromatin accessibility, excitatory activity, and cellular organization in L4. The data highlight *Thsd7a* as a novel gene involved in barrel organization and also implicate several other genes as novel regulators of laminar identity. The observation that barrel organization declines with age in wildtype mice is an interesting one that provides new avenues of research. Thus, overall, this work represents a significant contribution to our understanding of a complex cortical developmental process.

**Decision letter after peer review:**

Thank you for sending your article entitled "Cortical RORβ is required for layer 4 transcriptional identity and barrel integrity" for peer review at *eLife*. Your article has been reviewed by three peer reviewers, and the evaluation has been overseen by a Reviewing Editor and Catherine Dulac as the Senior Editor.

The three reviewers were split on the significance of the current study as written but all felt there were two findings in particular that, if they could be strengthened, would be of broad interest. These points were 1) the layer 4 to layer 5 "identity switch" of the RORb knockout neurons and 2) the degeneration of the organization of thalamocortical axons with age and/or *Thsd7* knockout. The reviewers raised concerns about interpretation of the data supporting each of these findings, and felt that additional experiments would be required to support either claim. These concerns are most clearly articulated in the comments of reviewer #3 below but were voiced by all three reviewers during the discussion period.

Essential revisions:

1) Alternative methods in addition to FACS purification by GFP and sequencing are needed to confirm that the layer 5 gene expression programs are indeed occurring in what would have been layer 4 RORb+ neurons.

2) Complementary methods in addition to vGLut2 staining are needed to characterize the arborization of TC axons in aging and in the *Thsd7* knockout mice to confirm that this organization are disrupted.

*Reviewer #1:*

This manuscript addresses the circuit functions of the layer 4 neuron marker RORb. The premise of the study is that layer 4 neurons are somewhat unique in their morphological arrangement into sensory input defined cellular aggregates and the authors presuppose that there is a transcriptional basis for this. Prior studies support this idea and already a number of transcription factors are known that link thalamocortical inputs with the organization of sensory barrels. Here the authors add RORb to this list, knocking it out and showing disruption of barrel formation. Global knockout of RORb was shown to reduce barrel formation (though barrels are still visible). Conditional knockout showed that RORb was required in the cortex (Emx1-cre) prior to barrel formation to have this effect. In both cases the effects get worse with age, though RORb knockout by CamKII-cre, after the barrels have formed, appeared to have no effect. The authors took advantage of the fact that the constitutive knockout allele expresses GFP to purify cells from S1. They compared RNA-seq data against Allen Brain in situ data for layer 4 markers and saw an apparently transformation of the cell fate in the KO neurons, with a down regulation of layer 4 markers and upregulation of layer 5. Using ATAC-seq to find the transcriptional mechanisms of these changes the authors found RORb motifs in many differentially-accessible regions comparing WT and KO, though not near the layer 4 genes. Many other sites contained binding sites for activity-sensing TFs, consistent with the known role of sensory input in barrel development. Finally the authors explore one potential target, *Thsd7a*, which also seems to disrupt some aspects of barrels.

The work is well done, the studies are rigorous, and the manuscript is well written. However it is not clear that the story is highly novel or significant. Several TFs (as the authors cite) are already known to couple barrel development with sensory input. The authors find some interesting information about RORb's specific functions in this process, but it is not clear that fundamental new insights emerge from the data. Overall I see the manuscript as being of great interest to specialists, but of limited interest to the broader audience.

*Reviewer #2:*

This study provides insights into the molecular mechanisms underlying RORβ's involvement in barrel formation and thus provides a tantalizing link across gene expression, chromatin accessibility, excitatory activity, and cellular organization in L4. The data highlight *Thsd7a* as a novel gene involved in barrel organization and also implicate several other genes as novel regulators of laminar identity. The observation that barrel organization declines with age in wildtype mice is an interesting one that provides new avenues of research. Thus, overall, this work represents a significant contribution to our understanding of a complex cortical developmental process. The body of the manuscript is well written, though I would urge the authors to carefully review the figure legends which contain errors and require some editing. In general, the conclusions drawn by the authors are supported by the data. The tests chosen for statistical analysis seem appropriate and sufficiently rigorous.

1) It would have been nice to see the progression from the molecular to the cellular level extended to the functional level to complete the whole picture, for instance by using a behavioral test of whisker function to see whether these molecular and cellular changes translate to a meaningful functional phenotype.

2) The mini EPSC frequency appears to stabilize in Rorb KO mice by P10, yet at P30 the authors observed changes in the expression of many activity-dependent TFs. To strengthen the claim that the changes in excitatory activity are consistent with the observed transcriptional changes, it would seem appropriate to perform the mini EPSC experiment at the P30 time point when most of the activity-dependent TFs show up-regulated expression.

3) It is unclear why normalization was only used when quantifying changes in barrel-septa contrast in the *Thsd7a* experiment (Figure 7). It seems prudent to use this approach in all the other experiments (Figures 1-3).

4) Figure 3D lacks an appropriate negative control for comparison at P60 (i.e. CamK2a-cre alone). This is especially important since the authors demonstrate an age-related desegregation of TCA.

*Reviewer #3:*

Using conditional KO strategies authors show that the transcription factor RORb is required in the cortex but not in thalamus for barrel formation; then, based on extensive genomic analyses of RORb+ neurons from layer 4 (L4) neurons , they find that RORb+ neurons loose some L4 molecular characteristics while acquiring new L5 characteristics. Finally they analyse 2 target genes of RORb looking for morphological change in barrel organisation.

Overall these data are interesting and well analysed; on one hand they confirm the role of RORb in L4 barrel formation ; this has had been proposed based on expression data and on gain of function experiments (Jabaudon et al., 2012). On the other hand, they bring an interesting controversy on the role of RORb in acquiring L4 molecular identity. L4 is generally considered to be part of the upper cortical plate neurons, and to share developmental origin and molecular identity with L2/L3 neurons (Oishi et al., 2016 ).

The following weak points need however to be addressed.

1) The most original claim of the study is that L4 neurons acquire L5 identity in *Rorb*-KO. However, this based solely on changes in gene expression, which are not really compelling, because effects are not consistent and vary strongly with age. To support their conclusion, authors need to provide further evidence on the laminar distribution of L4 and L5 /L6 molecular markers (e.g. Cux1, Brn2b, ctip2,...;). Additionally it would be important to know whether L4 neurons acquire new morphological characteristics, of L5 neurons such as pyramidal shape and sub-cerebral projections. Because GFP is expressed in the RORb deficient neurons these could be easily traced and analyzed.

2) The changes in gene expression in RORb KO vary most between P2 and P7: L5 markers (e.g. ctip2 and Fez1) are up regulated only at P2. Could there be some contamination of the P2 samples with L5 (in which RORb is also expressed), despite efforts in the dissection?

3) Another strong claim is that segregation of TCAs degrades with age. However, this is based exclusively on Vglut2 immunostaining with low resolution. This is questionable as the cortex matures, since Vglut2 staining becomes much more diffuse, possibly because of the arrival of other VGluT2^+^ cortical inputs than TCAs. In fact the difference between WT and RORb- KO becomes is less clear as animals age (Figure 2B) and authors note "loss of RORb did not significantly change the time course of TCA desegregation”. Thus without complementary approaches it is hard to make such strong conclusions. Previous studies showing desegregation of TCAs in the barrel cortex, secondary to cortex-specific deletions have used complementary methods such as tracing reconstructions of TCA axon terminals in the cortex (Ballester-Rosado, 2000, Lee et al., 2005).

4) For similar reasons as above, the phenotype of the barrel phenotype of *Thsd7*-KO is not really convincing. Although some higher resolution images are shown, these are not confocal, and would not allow rigorous measures of VGluT2^+^ terminals.

[Editors' note: further revisions were suggested prior to acceptance, as described below.]

Thank you for resubmitting your article "Cortical *Rorb* is required for layer 4 transcriptional identity and barrel integrity" for consideration by *eLife*. Your revised article has been reviewed by three peer reviewers, and the evaluation has been overseen by a Reviewing Editor and Catherine Dulac as the Senior Editor. The following individual involved in review of your submission has agreed to reveal their identity: Patricia Gaspar (Reviewer #3).

The reviewers have discussed the reviews with one another and the Reviewing Editor has drafted this decision to help you prepare a revised submission.

Summary:

This study provides insights into the molecular mechanisms underlying RORβ's involvement in barrel formation and thus provides a tantalizing link across gene expression, chromatin accessibility, excitatory activity, and cellular organization in L4. The data highlight *Thsd7a* as a novel gene involved in barrel organization and also implicate several other genes as novel regulators of laminar identity. The observation that barrel organization declines with age in wildtype mice is an interesting one that provides new avenues of research. Thus, overall, this work represents a significant contribution to our understanding of a complex cortical developmental process.

Essential revisions:

This is a revised manuscript, and in a previous round of review the reviewers requested a plan from the authors outlining how they would address what were seen as two major concerns. These were the claim from RNA-seq data that Layer 4 neurons were expressing Layer 5 gene programs, and the use of vGlut2 labeling to assess inputs to barrel cortex. The authors proposed to offer new data to address each of these points, using quantitative in situ to validate the RNA-seq and a viral tracing method to complement the vGLut2 staining evaluation of the organization of inputs in barrel cortex.

The authors did provide these data, however two of the reviewers felt that these new figures needed more clarification. In particular the way the authors presented and interpreted the viral input labeling experiment was a source of significant confusion. Comments from the reviewers on these two new datasets are below and these need to be addressed with text/figure revisions and/or tempering of claims.

Barrel cortex viral input labeling:

– The fact that the TC barrelless phenotype increases with age is not supported by the additional evidence provided. Furthermore, the quantification in Figure 2B actually shows the opposite trend: using measures of VMAT2 intensity in barrel hollows/septae the Ctrl/KO difference is most marked at P7 and P20, it seems less marked at P30 and it is no longer significant at P60. Therefore I would suggest not highlighting this in the Abstract and Discussion without better evidence.

– Figure 2—figure supplement 1 and Figure 7—figure supplement 1, compare TC tracing (very nice) with Vglut2 to show this is similar, but they do not illustrate/compare data from the ctrl and KO on the same or graph. The mean control values (n = ?) are represented by the dashed line; but then the SEM and stats need to be added. These 2 figures could be combined into one clearer figure.

– I do not understand the authors' interpretation of Figure 2—figure supplement 1C and Figure 7—figure supplement 1. First, what is the y-axis in Figure 2—figure supplement 1C and Figure 7—figure supplement 1 – barrel septal contrast of what? VGlut2? Comparing the absolute contrast levels of two methods of labeling inputs at a single time point or in a single genotype would not seem to mean much. I was assuming when they suggested this method that the authors would compare the Cherry signal in WT and KO to show it reproduced the lower signaling in the KO relative to WT like they saw with vGLut 2. I thought they would do a similar experiment and look at cherry labeling over time in the WT to show it decreases similar to the vGlut2 labeling. Those comparisons would seem to have been needed to address the concerns that were raised by the reviewers about the vGlut2 signal.

RNAscope in situ quantification:

– What is the time point for the in situ in Figure 5A? Why is the difference in Fezf2 expression so large in this sample whereas it is negligible at all time points in Supplementary Figure 4B? What did Tox look like in the sequencing data – it would be good to include it in Supplementary Figure 4B. The authors should also show the images for Tox in the supplementary figure because this is important to the validation. Finally, I do not understand what Figure 4E is trying to show or what the authors are concluding. There is no quantification so it is not possible to judge whatever conclusion was intended.

---

## [Author Response]

Essential revisions:1) Alternative methods in addition to FACS purification by GFP and sequencing are needed to confirm that the layer 5 gene expression programs are indeed occurring in what would have been layer 4 RORb+ neurons.2) Complementary methods in addition to vGLut2 staining are needed to characterize the arborization of TC axons in aging and in the Thsd7 knockout mice to confirm that this organization are disrupted.Reviewer #1:This manuscript addresses the circuit functions of the layer 4 neuron marker Rorb. The premise of the study is that layer 4 neurons are somewhat unique in their morphological arrangement into sensory input defined cellular aggregates and the authors presuppose that there is a transcriptional basis for this. Prior studies support this idea and already a number of transcription factors are known that link thalamocortical inputs with the organization of sensory barrels. Here the authors add RORb to this list, knocking it out and showing disruption of barrel formation. Global knockout of RORb was shown to reduce barrel formation (though barrels are still visible). Conditional knockout showed that RORb was required in the cortex (Emx1-cre) prior to barrel formation to have this effect. In both cases the effects get worse with age, though RORb knockout by CamKII-cre, after the barrels have formed, appeared to have no effect. The authors took advantage of the fact that the constitutive knockout allele expresses GFP to purify cells from S1. They compared RNA-seq data against Allen Brain in situ data for layer 4 markers and saw an apparently transformation of the cell fate in the KO neurons, with a down regulation of layer 4 markers and upregulation of layer 5. Using ATAC-seq to find the transcriptional mechanisms of these changes the authors found RORb motifs in many differentially-accessible regions comparing WT and KO, though not near the layer 4 genes. Many other sites contained binding sites for activity-sensing TFs, consistent with the known role of sensory input in barrel development. Finally the authors explore one potential target, Thsd7a, which also seems to disrupt some aspects of barrels.The work is well done, the studies are rigorous, and the manuscript is well written. However it is not clear that the story is highly novel or significant. Several TFs (as the authors cite) are already known to couple barrel development with sensory input. The authors find some interesting information about RORb's specific functions in this process, but it is not clear that fundamental new insights emerge from the data. Overall I see the manuscript as being of great interest to specialists, but of limited interest to the broader audience.

Several features of our study make it of broad interest. Prior studies identified transcription factors required for barrel development and laminar fate but stopped short of revealing the underlying transcriptional networks. Here, we begin to open the black box by which loss of a single transcription factor, through a cascade of changes in gene expression and chromatin accessibility, can shape activity-dependent cortical development. Our study dives deeper into the transcriptional regulation of barrel development than previous studies. The mechanisms we highlight have broader implications for understanding the complexity of transcriptional networks governing cellular identity and the diversity of transcriptional mechanisms altered by a single TF. Additionally, our description of TCA desegregation in adulthood and the increase in adult desegregation after *Thsd7a* KO, two findings further supported by additional VPM-specific experiments, opens up the barrel cortex to study mechanisms involved in age-related plasticity and cytoarchitecture maintenance.

Reviewer #2:This study provides insights into the molecular mechanisms underlying RORB's involvement in barrel formation and thus provides a tantalizing link across gene expression, chromatin accessibility, excitatory activity, and cellular organization in L4. The data highlight Thsd7a as a novel gene involved in barrel organization and also implicate several other genes as novel regulators of laminar identity. The observation that barrel organization declines with age in wildtype mice is an interesting one that provides new avenues of research. Thus, overall, this work represents a significant contribution to our understanding of a complex cortical developmental process. The body of the manuscript is well written, though I would urge the authors to carefully review the figure legends which contain errors and require some editing. In general, the conclusions drawn by the authors are supported by the data. The tests chosen for statistical analysis seem appropriate and sufficiently rigorous.1) It would have been nice to see the progression from the molecular to the cellular level extended to the functional level to complete the whole picture, for instance by using a behavioral test of whisker function to see whether these molecular and cellular changes translate to a meaningful functional phenotype.

While we agree extending the findings into behavioral readouts would be very interesting, those experiments are very time and resource intensive. We feel they are beyond the scope of this paper which focuses on molecular transcriptional mechanisms, and their roles in anatomical and physiological development.

2) The mini EPSC frequency appears to stabilize in Rorb KO mice by P10, yet at P30 the authors observed changes in the expression of many activity-dependent TFs. To strengthen the claim that the changes in excitatory activity are consistent with the observed transcriptional changes, it would seem appropriate to perform the mini EPSC experiment at the P30 time point when most of the activity-dependent TFs show up-regulated expression.

We previously performed mEPSC recordings at P19 and found that both frequency and amplitude returned to normal levels. We have added this data to Figure 6B.

3) It is unclear why normalization was only used when quantifying changes in barrel-septa contrast in the Thsd7a experiment (Figure 7). It seems prudent to use this approach in all the other experiments (Figures 1-3).

Normalization was not used in Figure 7D in the barrel-septa *contrast* calculation. It was used to measure *absolute* intensity within barrels or septa separately in Figure 7E-F. Because the contrast value is a ratio, it does not need to be normalized as both the barrels and septa would be corrected by the same measurement. We have added more detail regarding this aspect to the Materials and methods (subsection “Imaging and fluorescence quantification”).

4) Figure 3D lacks an appropriate negative control for comparison at P60 (i.e. CamK2a-cre alone). This is especially important since the authors demonstrate an age-related desegregation of TCA.

We are not able to detect a significant difference in barrel contrast between P30 and P60 in our control data (Figure 2B). There is a slight downward trend but the variance is too high and our N is too small to determine whether this is significant at the thresholds we’ve set. We do detect a significant change in TCA segregation comparing P7 to P20. We have added a more detailed description to the Results to make this distinction clearer (subsection “RORβ is required for postnatal barrel wall formation and influences segregation of thalamocortical afferents (TCAs)”).

Given that the *Rorb^f/f^* CamK2a-cre does not affect TCA organization, it is very unlikely that the CamK2a-cre alone will have an effect. We do not think excluding this control changes the interpretation of the negative result.

Reviewer #3:Using conditional KO strategies authors show that the transcription factor RORb is required in the cortex but not in thalamus for barrel formation; then, based on extensive genomic analyses of RORb+ neurons from layer 4 (L4) neurons , they find that RORb+ neurons loose some L4 molecular characteristics while acquiring new L5 characteristics. Finally they analyse 2 target genes of RORb looking for morphological change in barrel organisation.Overall these data are interesting and well analysed; on one hand they confirm the role of RORb in L4 barrel formation ; this has had been proposed based on expression data and on gain of function experiments (Jabaudon et al., 2012). On the other hand, they bring an interesting controversy on the role of RORb in acquiring L4 molecular identity. L4 is generally considered to be part of the upper cortical plate neurons, and to share developmental origin and molecular identity with L2/L3 neurons (Oishi et al., 2016 ).The following weak points need however to be addressed.1) The most original claim of the study is that L4 neurons acquire L5 identity in RORb-KO. However, this based solely on changes in gene expression, which are not really compelling, because effects are not consistent and vary strongly with age. To support their conclusion, authors need to provide further evidence on the laminar distribution of L4 and L5 /L6 molecular markers (e.g. Cux1, Brn2b, ctip2,...;). Additionally it would be important to know whether L4 neurons acquire new morphological characteristics, of L5 neurons such as pyramidal shape and sub-cerebral projections. Because GFP is expressed in the RORb deficient neurons these could be easily traced and analyzed.

We note that we are not claiming that loss of RORβ leads to a fate-switch in which all aspects of L4 identity are lost and all aspects of L5 identity are gained. Instead, we believe the data argue for a more nuanced view of “identity” comprised of multiple transcriptional circuits, only some of which are disrupted by loss of RORβ. This issue is likely in part due to a lack of clarity in our description. Our use of the phrase “shift in identity” understandably, but also unintentionally, evokes the concept of fate switching. We have edited the manuscript to make it clear (subsection “RORβ is required for expression of a layer 4 gene profile and repression of layer 5 genes”, Discussion) that we don’t think loss of RORb causes L4 neurons to take on a L5 identity. We have also added an analysis of additional layer-specific genes identified from the Allen Brain Atlas demonstrating that while many L4 and L5 genes are dysregulated, many are not (Figure 4—figure supplement 1E). Additionally, we don’t find any strong evidence that the L4 neurons have acquired L5 morphology. Instead, we propose that loss of RORb disrupts L4 specification, a process which appears to involve repression of L5 genes. In fact, we think RORb’s role is more likely in fine-tuning L4 identity after fate selection, and upregulation of deeper layer genes is a symptom of dysregulated L4 specification rather than a large-scale identity switch. We apologize for our lack of thoughtfulness in word choice and phrasing while describing these transcriptional changes.

2) The changes in gene expression in RORb KO vary most between P2 and P7: L5 markers (e.g. ctip2 and Fez1) are up regulated only at P2. Could there be some contamination of the P2 samples with L5 (in which RORb is also expressed), despite efforts in the dissection?

While we agree with reviewer #3 that age is a factor in upregulation of L5 genes, we do not agree with the assessment that P2 and P7 showed the most change between control and KO. The red line graphs of Figure 4B show that the younger ages show more variation in L5 genes changes with several genes down regulated at younger ages but ultimately upregulated at P30. The P30 adult time point shows the most consistent upregulation with the bulk of the L5 DEGs increased, and for several genes, also with the largest fold increase. We have added a line showing the mean LFC across the group of genes in Figure 4B to make this conclusion easier to assess. While Ctip2 does show the strongest upregulation at P2, FezF2 (Fez1 is not a DEG in our dataset) is only upregulated at P30.

The reviewer is insightful to suggest possible difficulty in the P2 microdissection. While we think the most compelling RNA-seq evidence is at P7 and P30 we performed RNAscope in situ hybridization to confirm upregulation of two L5 genes, FezF2 and Tox (Figure 4—figure supplement 1C-D). As an additional control, we analyzed expression changes in all of the genes differentially expressed between layers 4 and 5 of somatosensory cortex according to the Allen Brain Atlas (analysis and criteria in Figure 4—figure supplement 1). Most genes (74 L4 and 190 L5) did not, on average show a change between WT and *Rorb* KO. This would be unexpected if the changes seen in the genes shown in Figure 4 were due to contamination.

3) Another strong claim is that segregation of TCAs degrades with age. However, this is based exclusively on Vglut2 immunostaining with low resolution. This is questionable as the cortex matures, since Vglut2 staining becomes much more diffuse, possibly because of the arrival of other VGluT2^+^ cortical inputs than TCAs. In fact the difference between WT and RORb- KO becomes is less clear as animals age (Figure 2B) and authors note "loss of RORb did not significantly change the time course of TCA desegregation”. Thus without complementary approaches it is hard to make such strong conclusions. Previous studies showing desegregation of TCAs in the barrel cortex, secondary to cortex-specific deletions have used complementary methods such as tracing reconstructions of TCA axon terminals in the cortex (Ballester-Rosado, 2000, Lee et al., 2005).4) For similar reasons as above, the phenotype of the barrel phenotype of Thsd7-KO is not really convincing. Although some higher resolution images are shown, these are not confocal, and would not allow rigorous measures of VGluT2^+^ terminals.

To complement our VGLUT2 staining quantification and to address comments 3 and 4, we injected AAV carrying an mCherry reporter gene driven by the hsyn promoter (AAV-hSyn-mCherry) into VPM to anatomically label TCAs. We use VLGUT2 staining to define barrels and compared the barrel-septa contrast calculated from mCherry expression in VPM axons to that calculated from VGLUT2. Figure 2—figure supplement 1A-C and Figure 7—figure supplement 1 show that the contrast calculated from VPM-specific mCherry expressing axons was similar to the contrast calculated from VGLUT2 staining. This supports our interpretation that VPM TCA organization degrades with age and without functional *Thsd7a*, rather than as a result of the arrival of other VGLUT2^+^ cortical inputs as the cortex matures.

[Editors' note: further revisions were suggested prior to acceptance, as described below.]

Essential revisions:This is a revised manuscript, and in a previous round of review the reviewers requested a plan from the authors outlining how they would address what were seen as two major concerns. These were the claim from RNA-seq data that Layer 4 neurons were expressing Layer 5 gene programs, and the use of vGlut2 labeling to assess inputs to barrel cortex. The authors proposed to offer new data to address each of these points, using quantitative in situ to validate the RNA-seq and a viral tracing method to complement the vGLut2 staining evaluation of the organization of inputs in barrel cortex.The authors did provide these data, however two of the reviewers felt that these new figures needed more clarification. In particular the way the authors presented and interpreted the viral input labeling experiment was a source of significant confusion. Comments from the reviewers on these two new datasets are below and these need to be addressed with text/figure revisions and/or tempering of claims.Barrel cortex viral input labeling:– The fact that the TC barrelless phenotype increases with age is not supported by the additional evidence provided. Furthermore, the quantification in Figure 2B actually shows the opposite trend: using measures of VMAT2 intensity in barrel hollows/septae the Ctrl/KO difference is most marked at P7 and P20, it seems less marked at P30 and it is no longer significant at P60. Therefore I would suggest not highlighting this in the Abstract and Discussion without better evidence.

The paper does not argue that the effect of Rorb KO increases with age. Instead, we find that segregation of afferents declines with age and this effect is separate from the more profound loss of segregation seen in the knockout. This was highlighted by reviewer #2 in the prior set of reviews: “The observation that barrel organization declines with age in wildtype mice is an interesting one that provides new avenues of research. Thus, overall, this work represents a significant contribution to our understanding of a complex cortical developmental process.” To try to make this point clearer, we have altered the sentence in the Abstract which previously read “Interestingly, barrel organization also degrades with age.” to read: “Interestingly, barrel organization also degrades with age in wildtype mice.” We have also clarified this portion of the Discussion by changing: “Our observation that barrel organization declined with age is very interesting and possibly the first description of this phenomenon in mice (Rice, 1985)” to “Our observation that barrel organization declined with age in wildtype animals is very interesting and possibly the first description of this phenomenon in mice (Rice, 1985).” Please also note that we are careful to state in the Results that the change with age is common to both control and KO animals and that “while both age and loss of RORβ significantly reduced contrast, loss of RORβ did not significantly change the time course of TCA desegregation”.

– Figure 2—figure supplement 1 and Figure 7—figure supplement 1, compare TC tracing (very nice) with Vglut2 to show this is similar, but they do not illustrate/compare data from the ctrl and KO on the same or graph. The mean control values (n = ?) are represented by the dashed line; but then the SEM and stats need to be added. These 2 figures could be combined into one clearer figure.

The prior round of reviews included the following request from reviewer #3:

“3) Another strong claim is that segregation of TCAs degrades with age. However, this is based exclusively on Vglut2 immunostaining with low resolution. This is questionable as the cortex matures, since Vglut2 staining becomes much more diffuse, possibly because of the arrival of other VGluT2^+^ cortical inputs than TCAs…4) For similar reasons as above, the phenotype of the barrel phenotype of *Thsd7*-KO is not really convincing.”

I realize now that the reviewer may have thought that we were claiming that the Rorb KO phenotype increases with age (see point immediately above) but at the time we and the editors clearly interpreted the reviewer’s comments as referring solely to the aging phenotype, which is also present in wildtype mice, and the *Thsd7*-KO. Specifically, in the prior round of reviews Essential revision #2 reads “Complementary methods in addition to vGLut2 staining are needed to characterize the arborization of TC axons in aging and in the *Thsd7* knockout mice to confirm that this organization are disrupted.”

Based on our understanding of these requests we submitted a revision plan that included the a response as follows:

*“*To complement our vGLUT2 staining quantification we propose to inject anterograde tracers into the VPM of adult wild-type and *Ths7a* KO animals. We will then quantify the barrel-septa contrast both of vGLUT2 staining and the axonal label. We will also use high resolution confocal microscopy to document the presence of axonal terminals in the septa. If the loss of vGLUT2 contrast in the older adults and *Ths7a* KO is primarily due to loss of TCA localization to barrel hollows we should see a comparable loss of contrast in the VPM anterograde tracer. On the other hand, if the loss of vGLUT2 contrast is due to ingrowth of other vGluT2^+^ afferents, the TCA and vGLUT2 contrast will be mismatched and axonal labeling in the septa will be absent.”

We did carry out these experiments and illustrate them in Figure 2—figure supplement 1 and Figure 8—figure supplement 1. The number of animals for these experiments (4 for both figures) is stated in the legends. We did not propose to directly compare *Rorb* KO animals to wildtype animals or to produce a complete time course. While we did not produce a full time course for Figure 2—figure supplement 1, we replot the Vglut2 contrasts obtained for P7 for reference and for P30 to show that both measures of Vglut2 contrast are similar despite the experiments having been carried out many months apart. We felt it was important to show that the new experiments involving surgical and viral methods did not significantly alter Vglut2 contrast. The key result shown by these experiments is that TCAs specifically originating in the VPM do not show a difference in contrast compared to total Vglut2 staining. This data rules out the alternative interpretation of non-thalamic afferents raised by the reviewer in the first round of comments. If loss of Vglut2 contrast was due to the arrival of other VGluT2^+^ inputs we would not see such close alignment of Vlglut2 contrast and mCherry contrast at P30.

As the reviewer suggests, we have restated the N’s for the control data in the legends and have given the statistics in the two legends as well. We have also added the SD to the figures as requested.

We do not believe it would be clearer to combine the two figures into one since they come at very different points within the manuscript and address different questions. Figure 2—figure supplement 1 addresses the decline of barrel segregation with age in normal mice, while Figure 8—figure supplement 1 addresses the loss of barrel-septa contrast in the *Ths7a* KO.

– I do not understand the authors' interpretation of Figure 2—figure supplement 1C and Figure 7—figure supplement 1. First, what is the y-axis in Figure 2—figure supplement 1C and Figure 7—figure supplement 1 – barrel septal contrast of what? VGlut2? Comparing the absolute contrast levels of two methods of labeling inputs at a single time point or in a single genotype would not seem to mean much. I was assuming when they suggested this method that the authors would compare the Cherry signal in WT and KO to show it reproduced the lower signaling in the KO relative to WT like they saw with vGLut 2. I thought they would do a similar experiment and look at cherry labeling over time in the WT to show it decreases similar to the vGlut2 labeling. Those comparisons would seem to have been needed to address the concerns that were raised by the reviewers about the vGlut2 signal.

The y-axis in both figures compares the contrast of VGlut2 to the contrast of mCherry contained within VPM axons. Unlike intensity, contrast is not an absolute measure, but a relative one. It measures the fractional difference between labeling within the barrel compartment relative to that within the septal compartment as a measure of the segregation of axons. A direct comparison of the spatial contrast (segregation) between Vglut2 labelled axons and mCherry-labelled axons in the same tissue was precisely the point of this experiment. What this figure shows clearly is that both methods of labeling axons produce a nearly identical measure of contrast. In other words, the original concern that “Vglut2 staining becomes much more diffuse, possibly because of the arrival of other VGluT2^+^ cortical inputs than TCAs” can be ruled out.

RNAscope in situ quantification:– What is the time point for the in situ in Figure 5A?

The in situ was performed at P30 which is stated in the Materials and methods. We have now added this to the legend for clarity.

Why is the difference in Fezf2 expression so large in this sample whereas it is negligible at all time points in Supplementary Figure 4B?

The scale in Supplementary Figure 4B made the difference appear negligible because Fezf2 levels decline greatly by P30. We have replotted the data from Supplementary Figure 4B so that the close agreement between the sequencing and in situ can be better appreciated. We have also included this panel in a new main figure (Figure 5) as suggested below.

What did Tox look like in the sequencing data – it would be good to include it in Supplementary Figure 4B. The authors should also show the images for Tox in the supplementary figure because this is important to the validation.

We have now included both images for *Tox* and sequencing data for *Tox* to allow appreciation of the correspondence. This now comprises a new Figure 5—figure supplement 1.

Finally, I do not understand what Supplementary Figure 4E is trying to show or what the authors are concluding. There is no quantification so it is not possible to judge whatever conclusion was intended.

The former Supplementary Figure 4E supports the point that not all L4-specific and L5-specific genes are affected by the knockout, which one would expect if the change we observe were simply due to L5 contamination (a concern raised by reviewers in the first round of comments). This panel is now moved to Main Figure 4E. We have added a statistical analysis (Fisher exact test) to show that although only a subset of L4 and L5 genes are affected, downregulation of L4 genes and upregulation of L5 genes are highly overrepresented (Figure 4F). As explained in the legend and the text, these include all of the genes with 1.5 L4:L5 fold-change and expression > 1.6 as identified by the Allen Brain Atlas differential search. Since these genes exclude the 31 L4 genes and 50 L5 differentially expressed genes shown in Figure 4A-C, they are by definition, not statistically altered by the knockout, as shown by the similarity of the mean fold-change (black line) to the gray line indicating no change. Therefore, this new analysis aims to make two points: (1) the data are not consistent with simple contamination of Rorb KO samples by L5 cells, and (2) provide additional statistical support for the conclusion that L4 and L5 genes are significantly altered. We have added clarification of these results to the legend and to the Results subsection “RORβ is required for expression of a layer 4 gene profile and repression of layer 5 genes”.